# FROM PERCEPTION TO PUNCHLINE: EMPOWERING VLM WITH THE ART OF IN-THE-WILD MEME

## ABSTRACT

Generating humorous memes is a challenging multimodal task that moves beyond direct image-to-caption supervision. It requires a nuanced reasoning over visual content, contextual cues, and subjective humor. To bridge this gap between visual perception and humorous punchline creation, we propose *HUMOR*, a novel framework that guides VLMs through hierarchical reasoning and aligns them with group-wise human-like preferences. First, HUMOR employs a hierarchical, multi-path Chain-of-Thought (CoT): the model begins by identifying a template-level intent, then explores diverse reasoning paths under different contexts, and finally anchors onto a high-quality, context-specific path. This CoT supervision, which traces back from ground-truth captions, enhances reasoning diversity. We further analyze that this multi-path exploration with anchoring maintains a high expected humor quality, under the practical condition that high-quality paths retain significant probability mass. Second, to capture subjective humor, we train a pairwise reward model that operates within groups of memes sharing the same template. Following established theory, this approach ensures a consistent and robust proxy for human preference, even with noisy labels. The reward model then enables a group-wise reinforcement learning optimization, guaranteeing that the model's humor quality does not degrade beyond a bounded amount. Experiments show that HUMOR empowers various base VLMs with superior reasoning diversity, more reliable preference alignment, and higher overall meme quality compared to strong baselines. Beyond memes, our work presents a general training paradigm for open-ended, human-aligned multimodal generation, where success is guided by comparative judgment within coherent output groups.

## 1 INTRODUCTION

Creativity in multimodal generation increasingly moves beyond literal description to subjective and context-dependent outputs, such as humor, aesthetics, style, and social alignment, where quality is not defined by a single ground-truth but instead guided by human preference (Yadav et al., 2025; Burn & Kress, 2018). While recent vision–language models (VLMs) achieve strong results on captioning and visual question answering (Kuang et al., 2025; Ghandi et al., 2023), these tasks still admit relatively objective targets (Yan et al., 2023), leaving open how to train systems for goals that are open-ended and preference-driven (Bhatia et al., 2024). Current approaches often model meme generation as a direct image-to-caption task optimized with a fixed loss. This collapses the reasoning process into the decoder, suppresses intermediate interpretation, and tends to produce captions that are fluent yet shallow or not humorous (Yadav et al., 2025).

Meme generation provides a demanding testbed for this challenge. To succeed, a model must identify a template's latent intent, ground it in context-specific details of the image (objects, expressions, layout), and produce a caption that completes a metaphor or subverts expectation in a way humans find funny. This requires both **hierarchical reasoning** and **alignment with subjective humor**. Prior work typically uses text-only humor cues or global regression-style funniness scores (Baluja, 2024; Kalloniatis & Adamidis, 2024; Zhu et al., 2025a), assuming humor is directly comparable across templates. In practice, however, human judgments are more reliable within a group of memes that share the same template or theme, and far less stable across groups with different conventions. Ignoring this structure introduces noise, harms generalization, and encourages shortcuts that reward superficial overlap instead of genuine humor fit.

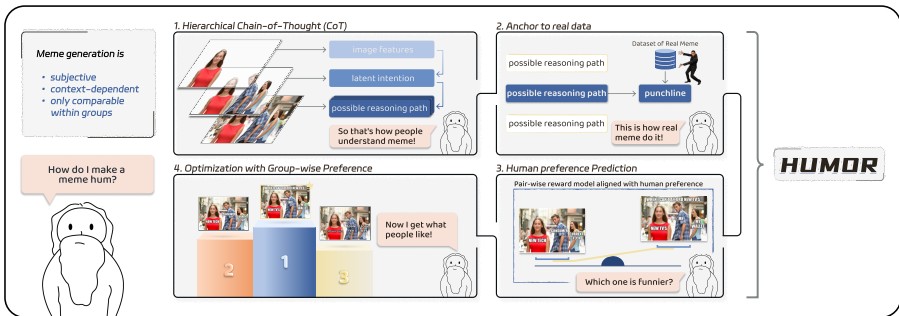

Figure 1: Overview of the HUMOR framework. Given a template image, it first performs hierarchical reasoning with a multi-path CoT: a template-level stage infers latent intent, and a context-level stage explores multiple paths grounded in visual content. One high-quality path is anchored by tracing back from ground-truth captions, supporting diversity while ensuring a conditional humor lower bound. A pairwise reward model then compares memes only within groups sharing the same template, maintaining rank consistency and providing a proxy signal of human-like preference. This reward enables group-wise RL to update the generation model in a stable way, ensuring expected humor does not degrade. Together, these components show how HUMOR combines structured reasoning, group-wise preference modeling, and stable optimization for meme generation.

A second limitation is the lack of an explicit reasoning-then-realization view. Directly sampling captions from images removes control over the interpretive process and makes it difficult to steer generation. Recent evidence shows that chain-of-thought (CoT) intermediates improve reasoning in VLMs. We argue that meme generation requires not just a single trace but a **hierarchical, multi-path reasoning process**: a template-level stage that infers canonical intent, followed by a context-level stage that grounds the intent in specific visual details. Different reasoning paths may lead to distinct metaphor bindings or punchlines. Exploring multiple paths and then anchoring one path with ground-truth data ensures diversity while, as our analysis shows, preserving a conditional lower bound on expected humor whenever high-quality paths keep a meaningful share of probability and the remaining paths are not much worse. Meeting these conditions requires optimizing generation toward human-preferred humor. Since humor cannot be directly measured, we design a **pairwise reward model** that maintains rank consistency within groups and prove that it inherits theoretical guarantees. This model provides a stable proxy signal of human-like preference, and further enables group-wise RL to ensure that expected humor cannot degrade beyond a bounded amount.

Figure 1 provides a high-level overview of *HUMOR*. It illustrates the main challenges in meme generation and how our framework addresses them: hierarchical reasoning with multi-path CoT, group-wise preference modeling, and stable optimization via RL. Taken together, these insights motivate our framework **HUMOR**: Hierarchical Understanding and Meme Optimization via Reinforcement learning. *HUMOR* separates reasoning from realization, respects group-wise comparability, and turns preference signals into stable policy updates. In summary, our contributions are:

1. **A new formulation of meme generation** as an open-ended, group-wise reasoning problem, together with a hierarchical multi-path CoT supervision scheme that separates template-level intent from context-level grounding. This framing exposes interpretable reasoning traces and lays the foundation for preference optimization.
2. **Theoretical analysis** showing that multi-path CoT supervision preserves a conditional humor lower bound and preference learning ensures consistent within-group ordering with provable stability. These results not only explain why our approach remains robust under noisy and subjective labels, also provide transferable insights for other open-ended, human-aligned generation tasks.
3. **Comprehensive experiments** across multiple base models showing that *HUMOR* improves reasoning diversity, preference alignment, and overall meme quality.

## 2 RELATED WORK

### 2.1 EVOLUTION OF VISION-LANGUAGE MODELS FOR MULTI-MODAL PROCESS

The pursuit of unified vision-language modeling has progressed through three distinct phases of architectural innovation. Early foundational work established bidirectional frameworks for cross-

modal understanding: ERNIE-ViLG (Zhang et al., 2021) and the Unifying Multi-modal Transformer (Huang et al., 2021) pioneered transformer-based architectures that jointly optimized text-to-image and image-to-text generation through multi-modal tokenization and autoregressive objectives. Concurrently, Zero-Shot Text-to-Image Generation (Ramesh et al., 2021) demonstrated the scalability potential of such approaches through their zero-shot text-to-image generation framework, establishing critical baselines for large-scale multi-modal pretraining.

Contemporary breakthroughs have redefined architectural paradigms through multimodal unification. Models like Show-o (Xie et al., 2024) and MonoFormer (Zhao et al., 2024) successfully fused autoregressive and diffusion mechanisms within singular architectures via shared attention layers. Beyond architectural fusion, recent research highlights the critical role of reasoning strategies. Chain-of-Thought (CoT) prompting has been empirically shown to enhance the complex reasoning capabilities of VLMs by eliciting intermediate rationales (Zhang et al., 2023; Hu et al., 2024). Building upon these advancements, our work leverages multi-modal comprehension capabilities to address the unique challenges of meme generation—particularly its requirement for hierarchical reasoning and understanding subjective humor.

## 2.2 MEME ANALYSIS, GENERATION, AND ALIGNMENT

**Humor Analysis and Generation.** Computational humor draws from established linguistic and anthropological theories (Apte, 1985; Binsted et al., 2006) to formally model incongruity and semantic shifts. Internet memes have emerged as a vital component of digital culture, prompting substantial scholarly attention to their multi-modal communications. Extensive research has focused on analyzing topics (Du et al., 2020), semantics (Xu et al., 2022), and emotions (Sharma et al., 2020) conveyed in memes. The evolution of meme generation techniques has progressed through distinct technological phases. Initial systems employed rule-based architectures, exemplified by Oliveira et al. (2016)'s template-driven approach using standardized structures like *"One does not simply X"*, and Wang & Wen (2015)'s dual-channel model integrating textual and visual features. The advent of deep learning catalyzed more sophisticated paradigms. Peirson and Tolunay pioneered this transition with Dank Learning (Peirson V & Tolunay, 2018), combining Inception V3 image encoders with attention-enhanced LSTM decoders. Subsequent innovations introduced transformer architectures: Sadasivam et al.'s MemeBot (Sadasivam et al., 2020) and Vyalla et al.'s Memeify (Vyalla & Udandarao, 2020) demonstrated enhanced text-image alignment through multi-modal fusion.

Recent breakthroughs leverage large language models (LLMs) and VLMs to achieve unprecedented scale. Memecraft (Wang & Lee, 2024) enables targeted meme creation for social advocacy. Addressing multi-image complexity, Chen et al. proposed XMeCap (Chen et al., 2024b), introducing a two-stage framework with supervised fine-tuning guided by novel similarity metrics. Concurrently, benchmark datasets have emerged to evaluate capabilities. MemeCap (Hwang & Shwartz, 2023) provides metaphor annotations, while the New Yorker benchmarks series (Hessel et al., 2023b;a) assess humor comprehension through caption matching and explanation tasks. The MCC dataset (Sharma et al., 2023) further incorporates external knowledge for abstraction analysis.

While capability has scaled, aligning models with *subjective* human preferences remains a critical frontier. Unlike objective tasks with ground-truth, humor and creativity require modeling diverse and often noisy judgments. Recent works have begun to address this by aligning models with diverse human values (Zhou et al., 2024) and exploring personalized or pluralistic strategies (Feng et al., 2024). Specifically in the domain of humor, Song et al. (2025) highlight the challenges of modeling subjective humor preferences using LLMs. Our work advances this direction by proposing a group-wise preference formulation, mitigating the noise inherent in cross-context humor comparison.

## 3 PROBLEM FORMULATION

This section formulates the core assumptions and components used throughout the paper. We begin by defining the structured meme space and the principle of group-wise comparability. Subsequently, we characterize the local pairwise preference data and posit the existence of a latent humor functional within each group. An observation model linking latent humor to pairwise comparisons is then introduced. Finally, we establish the objective for a meme generator, defining the key evaluation quantities. The result is a self-contained problem formulation that highlights group-wise comparability while remaining agnostic to specific training methodologies.

**Meme Space and Group-wise Comparability:** Let $\mathcal{M}$ denote the set of all memes under consideration. Each meme is represented as a multimodal pair $m = (I, c)$, where $I \in \mathcal{I}$ is a base image and $c$ is a textual punchline rendered at designated positions. Many memes are created from widely shared *templates* and are interpreted through context-dependent associations. Since humor is highly subjective and context-sensitive, absolute comparisons of humor across different templates are often ill-posed. Therefore, we assume and partition the meme space into $K$ disjoint groups:

$$\mathcal{G} = \{G_1, \ldots, G_K\}, \qquad G_k \subset \mathcal{M}, \qquad G_k \cap G_\ell = \emptyset \ (k \neq \ell),$$

Memes within the same group share a comparable structure (e.g., the same template, or punchline schema), which enables meaningful humor comparison. We posit that human judgments of humor are reliable *within* each group $G_k \in \mathcal{G}$, but do not assume comparability *across* different groups.

**Local Preference Data:** For a given group $G$, we collect human annotations indicating which of two memes is considered funnier. Formally, for $m_i, m_j \in G$, define $y_{ij}^G = \mathbb{I}[m_i \succ m_j] \in \{0, 1\}$ where $m_i \succ m_j$ denotes a local preference that $m_i$ is judged to be funnier than $m_j$. The dataset consists of triples $(G, (m_i, m_j), y_{ij}^G)$ sampled from a pairing distribution over $G$. We allow for incompleteness (not all pairs are labeled) and noise (due to inter-annotator disagreement). We adopt two mild yet standard assumptions from preference learning Christiano et al. (2023): (i) *local comparability*: preferences are elicited and interpreted only within a fixed group $G$; (ii) *weak transitivity*: in expectation, if $m_i \succ m_j$ and $m_j \succ m_\ell$, then $m_i \succ m_\ell$ is more likely than its reversal, without requiring a strict total order.

**Latent Humor within A Group:** Within each group $G$, we posit the existence of a latent humor functional $h_G : G \to [0, 1]$, This functional maps each meme $m \in G$ to a scalar reflecting its relative likelihood of being judged as funny by humans in that group. We do not assume that $h_G$ is calibrated across different groups, nor that $h_G$ and $h_{G'}$ are directly comparable when $G \neq G'$.

**Observation Model for Pairwise Labels:** Pairwise comparison labels are modeled as noisy observations of underlying differences in latent humor. Formally, we assume:

$$\Pr[m_i \succ m_j \mid G] = \Lambda(h_G(m_i) - h_G(m_j)), \tag{1}$$

where $\Lambda : \mathbb{R} \to (0, 1)$ is a strictly increasing link function (e.g., logistic or probit) (Sun et al., 2025). Eq. 1 captures the intuition that the probability of preferring $m_i$ to $m_j$ depends *only* on their latent humor gap within the same group: when $h_G(m_i) \approx h_G(m_j)$, the choice is nearly ambiguous (probability $\approx 1/2$); as the gap increases, the probability moves smoothly toward 1 (if $h_G(m_i) > h_G(m_j)$) or 0 (otherwise), capturing that larger humor gaps lead to more consistent comparisons.

**Generation Goal and Evaluation Quantities:** A meme generation model is defined as a conditional probability distribution over punchlines (or called captions) given an image: $\pi_\theta(\cdot \mid I) : I \mapsto \Delta(\mathcal{C})$, where $\Delta(\mathcal{C})$ denotes the set of probability distributions over the caption space $\mathcal{C}$. A meme sample $m = (I, c)$ is instantiated by sampling a caption $c \sim \pi_\theta(\cdot \mid I)$. For any target group $G$ containing meme candidates derived from the base image $I$, the expected within-group humor of $\pi_\theta$ is defined as $\mathcal{H}_G(\theta) = \mathbb{E}_{c \sim \pi_\theta(\cdot|I)}[h_G((I, c))]$. The overall population objective is then obtained by aggregating over groups according to a task-specific distribution over $(I, G)$:

$$\mathcal{H}(\theta) = \mathbb{E}_{(I,G)}[\mathcal{H}_G(\theta)]. \tag{2}$$

## 4 HUMOR FRAMEWORK

We propose **HUMOR**: **H**ierarchical **U**nderstanding and **M**eme **O**ptimization, a framework that guides VLMs through hierarchical reasoning and aligns them with group-wise humor preferences. The overall process of the framework is shown in Fig. 3. HUMOR consists of three integrated components: hierarchical CoT supervision, pairwise reward modeling, and group-wise policy optimization. These components collectively ensure diverse reasoning, consistent preference learning, and stable humor improvement. Propositions 1, 2, C.3, and 4 formally establish the coherence and controllability of the overall framework.

## 4.1 HIERARCHICAL CHAIN-OF-THOUGHT SUPERVISION

Meme creation mirrors a hierarchical cognitive routine: humans first parse what a visual template affords, and then realize a chosen intent with text that fits the surrounding context Flamson & Barrett (2008). We therefore model meme generation as a two-stage reasoning process, separating (i) intent inference from the image and (ii) context-sensitive textual realization of that intent. In practice, however, training trajectories are often single-path because they are derived from a single gold caption: back-deriving a rationale from one answer yields only one route.

As shown in Fig. 13,When trained with the final meme answer as a single path, the model collapses the reasoning process into a single decoding step, failing to develop true association and in-depth understanding. It only establishes a superficial mapping from user input to the current answer, leading to superficial captions and inability to adapt to the template nature of memes. Therefore, it is necessary to first explore the template's latent intent and core characteristics, and deliberately generate multiple semantic association possibilities under this template during reasoning to support the flexible use of the template's high-level meaning. To address this, we conceptualize the meme understanding and reasoning process as a hierarchical chain-of-thought $r = (r_{\text{tmpl}}, r_{\text{cont}})$, which explicitly decouples template-level interpretation from context-level grounding. Captions are then realized by sampling from $P_\phi(c \mid r, I)$.

To approximate human authorship, we supervise CoT in two stages. The process is shown in Fig. 2. In Stage 1, we first train the model $P_\phi(r \mid I, \hat{U})$ with *multi-path* reasoning traces synthesized by auxiliary LLM "teachers" under $(I, \hat{U})$, where $\hat{U}$ is a candidate set of *potential user contexts* (e.g., emotions, intentions, scenarios) suggested by the template's affordances (Appendix B). At inference, the model explores multiple reasoning paths conditioned only on $I$, while hypothesizing a candidate set of *potential user contexts* $\hat{U}$ (e.g., emotions, intentions, or scenarios a user might want to express). Concretely, the model generates reasoning candidates (multiple associative scenarios) $\{r^{(i)}\}_{i=1}^N \sim P_\phi(r \mid I, \hat{U})$, encouraging broad coverage of diverse interpretations. This part is similar to how humans brainstorm several possible jokes before finalizing one.

In Stage 2, when groundtruth captions are available, we anchor one path $\tilde{r}$ to be consistent with the punchlines of real memes (i.e., ground-truth captions) by incorporating the *actual user context $U$*, which is inferred from ground-truth captions. Formally, we select $\tilde{r} = \arg\max_r P_\phi(c \mid r, I, U)$, which ensures trajectory consistency while preserving the diversity acquired in Stage 1. At inference time (no gold caption), the generator ranks and selects among Stage 1 paths using its internal scoring/decoding policy (see Appendix B for construction details and examples).

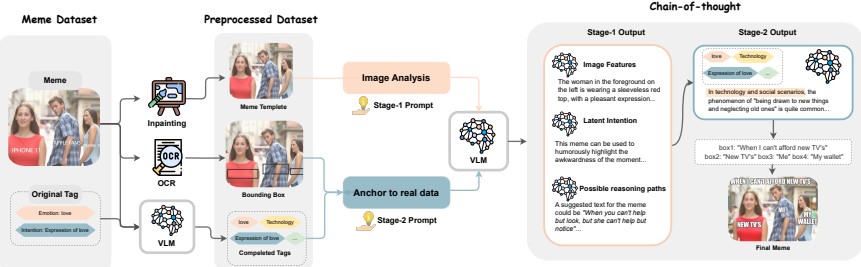

Figure 2: This diagram shows the dataflow for constructing hierarchical CoT supervisions. Stage 1 explores multiple reasoning paths that bind a template to different context-specific details. Stage 2 anchors one high-quality path from ground-truth, preserving diversity while preventing collapse.

The benefit of this design can be formalized as follows. Let $\tilde{h}_G : \mathcal{R} \to [0, 1]$ denote group-relative humor measure defined over reasoning paths. Suppose there exists a set of "star" paths (i.e., better paths) $R^\star$ with probability mass $\alpha > 0$ under the reasoning distribution, and the average humor gap between non-star paths and the best paths is bounded by $\delta \geq 0$. Then, we have the following guarantee:

**Proposition 1** (Conditional humor lower bound). *Normalizing* $\max \tilde{h}_G = 1$, *the expected humor after two stages CoT supervision satisfies:*

$$\mathbb{E}_{r \sim P_\theta}[\tilde{h}_G(r)] \geq 1 - (1 - \alpha)\delta.$$

Intuitively, as long as promising reasoning paths retain non-negligible probability ($\alpha$ is not too small) and the remaining paths are only mildly worse (small $\delta$), the process of exploration and anchoring preserve a nontrivial lower bound on expected humor. Conversely, Stage 1 exploration sustains multi-hypothesis diversity—preventing entropy from collapsing toward zero in the no-exploration limit—while Stage 2 anchoring ensures that a non-negligible portion of probability mass is concentrated on promising paths. Thus, Our proposed CoT framework broadens the breadth of interpretations without sacrificing quality. However, while $\alpha$ is naturally ensured by anchoring toward ground-truth paths, the humor gap $\delta$ remains uncontrolled: some generated paths may still be substantially less funny than others. To minimize $\delta$, we need an additional mechanism that reflects human humor preferences and can guide optimization beyond imitation.

## 4.2 REWARD MODELING FROM PAIRWISE PREFERENCES

The ideal learning objective would be to recover the latent humor function $h_G(m)$ for each meme $m$. Since humor is inherently subjective and lacks a global scale, this is infeasible in practice. We therefore adopt an *order-consistent* view of reward modeling (following established theory (Sun et al., 2025)) and instantiate it in our *group-wise* meme setting. Under this formulation, the reward serves as a *within-group surrogate* of $h_G$, trained only from relative judgments, avoiding ill-posed cross-group calibration. Intuitively, hierarchical CoT ensures that high-quality paths retain a meaningful probability mass (the $\alpha$ condition via Stage 2 anchoring), while the reward model supplies the preference signal necessary to *shrink the average gap among plausible paths* (addressing the $\delta$ condition). This transforms open-ended exploration into a tractable selection problem.

Each meme $m = (I, c)$ is encoded to a feature vector $\Psi(m) \in \mathbb{R}^d$ using a VLM as the encoder. Let a scoring head $f_\phi : \mathbb{R}^d \to \mathbb{R}$ map this feature vector $\Psi(m)$ to a scalar score. we denote this score as $s_\phi(m) = f_\phi(\Psi(m))$. For any pair of memes $(m_i, m_j)$ from the same group $G$, we define the predicted preference probability as:

$$\widehat{p}_{ij}^G = \sigma\big(s_\phi(m_i) - s_\phi(m_j)\big), \tag{3}$$

where $\sigma(\cdot)$ denotes the logistic function; The model is trained by minimizing the binary cross-entropy over human-annotated or auto-generated preference pairs.

Building upon the reward modeling formulation in Eq. 3, we now formalize two key theoretical properties (order consistency and stability) that justify its use in our within-group meme setting.

**Proposition 2** (Rank consistency). *Under the observation model of Eq. 1 with any strictly increasing link function, minimizing the pairwise preference loss recovers the same within-group ordering as the latent humor function $h_G$. Complete proofs are provided in Appendix C.*

**Proposition 3** (Robustness to label noise (margin-aware)). *Let $\Delta_{ij}^G = h_G(m_i) - h_G(m_j)$ denote the true humor gap, and assume the annotation process has pairwise error rate $\varepsilon$. For pairs satisfying $|\Delta_{ij}^G| \geq \delta$, the probability of order reversal is bounded above by a function decreasing in $\delta$ and increasing in $\varepsilon$; large humor gaps are therefore preserved even under noisy labels.*

These propositions, while grounded in the order-consistent analysis of Sun et al. (2025), are specifically instantiated under our group-wise comparability. They serve as the theoretical *drivers* to reduce the humor gap $\delta$ after CoT has secured $\alpha$. Since pairwise data can be sparse, we aggregate $\widehat{p}_{ij}^G$ into a coherent within-group ranking via *Expected Borda Count (EBC)* (see Appendix G for more explanations and implementations). For a candidate set $\mathcal{S}_G$, each meme's EBC score equals its expected number of wins against others under the model in Eq. 3. This provides a stable training target, and inherits expected order consistency when the pairwise model is consistent (Appendix C). Detailed constructs and examples of pairwise data are provided in Appendix E.

## 4.3 GROUP-WISE POLICY OPTIMIZATION

Following the CoT supervision stage and reward model training, we further fine-tune the meme generator to *increase* the probability of higher-ranked captions. Concretely, we leverage the trained reward model and adopt a **Group-wise Relative Policy Optimization (GRPO)** objective Shao et al. (2024). For a candidate set of memes $\mathcal{S}_G$ with ranking $q_G$ from EBC, the reinforcement fine-tuning loss is:

$$\mathcal{L}_{\text{GRPO}}(\theta) = \mathbb{E}_{(I,G)}\Big[ -\sum_{m_k \in \mathcal{S}_G} q_G(m_k) \log \pi_\theta(c_k \mid I)\Big] + \beta\, \mathbb{E}_I\big[\text{KL}(\pi_\theta(\cdot \mid I) \,\|\, \pi_{\text{ref}}(\cdot \mid I))\big], \tag{4}$$

where $\pi_{\text{ref}}$ denotes the policy obtained after CoT training. The first term aligns $\pi_\theta$ with the group-local preference distribution $q_G$ (rank-consistent with $h_G$). Since prior perference optimization analyses Christiano et al. (2023); Neu & Szepesvari (2012); Haarnoja et al. (2018) often propose optimistic lower bounds (second term), we also adopt a corrected, KL-controlled guarantee that holds under our setting and noise model. Specifically, the original upper bound on the humor-score deviation (induced by preference noise) can be refined under GRPO into a bound that scales with the KL between the trained policy and the reference policy (proof in Appendix D).

**Proposition 4** (Bounded change of expected humor under GRPO)**.** *Assume Proposition 2 holds and* $h_G \in [0, 1]$. *Let* $\Delta_{\text{KL}} = \mathbb{E}_I[\text{KL}(\pi_\theta(\cdot \mid I) \,\|\, \pi_{\text{ref}}(\cdot \mid I))]$. *Then*

$$\mathbb{E}_{(I,G)}\Big[\mathbb{E}_{c \sim \pi_\theta(\cdot|I)}\, h_G((I, c))\Big] \;\geq\; \mathbb{E}_{(I,G)}\Big[\mathbb{E}_{c \sim \pi_{ref}(\cdot|I)}\, h_G((I, c))\Big] \;-\; \sqrt{\tfrac{1}{2}\,\Delta_{\text{KL}}}.$$

*Hence, if GRPO enforces* $\Delta_{\text{KL}} \leq \tau$, *the expected humor cannot drop by more than* $\sqrt{\tau/2}$; *with the first term pull toward* $q_G$, *this ensures non-decreasing behavior within a bounded KL neighborhood.*

This bound, derived via Pinsker's inequality, formalizes the stability underlying our approach in practice: CoT supervision supplies sufficient support ($\alpha$), the reward model and EBC induce a group-local order that reduces $\delta$, and GRPO turns this order into controlled policy updates. In sum, our use of order-consistent surrogates aligns with established theory, but the *group-wise instantiation*, the *corrected KL-based bound*, and the *integration with multi-path CoT for open-ended generation* are key ingredients that make the approach effective and verifiable for meme generation.

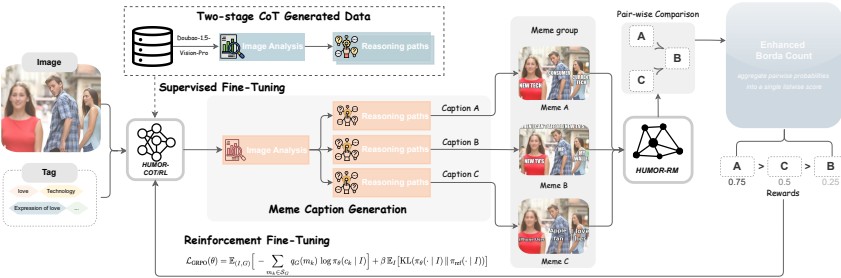

Figure 3: Training Pipeline of HUMOR. Multi-path CoT expands reasoning coverage and anchors a canonical path; the reward model translates pair data into a rank-consistent group-level signal (via EBC); GRPO then updates the generator toward higher-ranked captions.

## 5 EXPERIMENT

### 5.1 MEME QUALITY AND DIVERSITY WITH HUMOR

**Settings:** We evaluate the proposed HUMOR framework against several competitive baselines and model variants. Concretely, the compared systems include multiple open-source and closed-source VLMs, as well as our *HUMOR-CoT* model, which is fine-tuned with the hierarchical CoT design. To further investigate the efficacy of alternative CoT methods for meme generation, we also include several advanced CoT frameworks (Kim et al., 2023; Chen et al., 2024a), all trained under the same data and protocol. See Appendix H.1 for detailed training settings and Appendix A for the details of datasets. Given the highly open-ended and human-aligned nature of meme generation, we prioritize human evaluation. Human annotators are asked to assign scores to generated memes along four predefined quality axes. In addition, we adopt the conventional metric of text-level similarity between generated captions and their original reference texts. To further quantify generation diversity, we introduce a novel metric called **Distance under Context Swap**. This measure replaces the original context in the training set with a randomly selected one (kept consistent across models), and computes the textual divergence between the resulting caption and the original. A larger distance suggests reduced overfitting to SFT labels and better adaptability to new contexts. Due to observed instability in VLM-based rubric scores for meme evaluation (Sec. 5.2), we incorporate only one VLM-based metric: a human-likeness score. This is formulated as a binary classification estimating the probability that a meme was created by a human, with higher values indicating better. We adopt Gemini-2.5-pro as the evaluator for computing Human Rate, as it demonstrates the most stable and consistent behavior among candidate VLMs in our evaluator reliability analysis (Appendix L). For a more detailed description of the indicator meanings and evaluation criteria, see Appendix I.1.

Table 1: **Evaluation results across open-source models, closed-source models, and Qwen2.5-7B-Instruct fine-tuned with our proposed and different CoT methods.** Metrics include context-swap distance (diversity criterion), text-level similarity (sim. to original meme text), human evaluation (Humor, Readability, Relevance, Originality), and Human Rate.

| Category / Model | Human Evaluation (0-5) ↑ | | | | Text-level Similarity ↑ | Context-swap Distance ↑ | Human Rate (%)↑ |
|---|---|---|---|---|---|---|---|
| | Humor | Readability | Relevance | Originality | | | |
| **Open-source Models** | | | | | | | |
| Qwen2.5-7B-Instruct (Bai et al., 2025) | 2.39 | 3.35 | 2.91 | 2.57 | 0.576 | 0.564 | 75.7 |
| Qwen2.5-32B-Instruct (Bai et al., 2025) | 2.54 | 3.52 | 3.09 | 2.76 | 0.564 | 0.566 | 82.2 |
| InternVL3-8B (Zhu et al., 2025b) | 2.39 | 2.79 | 3.04 | 2.79 | 0.545 | 0.564 | 62.7 |
| GLM-4.1V-9B-Thinking (Hong et al., 2025) | 1.73 | 2.62 | 2.75 | 2.71 | 0.602 | 0.572 | 45.1 |
| Keye-VL-8B-preview (Team et al., 2025) | 2.35 | 3.19 | 2.99 | 2.71 | 0.585 | 0.580 | 69.0 |
| **Closed-source Models** | | | | | | | |
| GPT-4o (OpenAI, 2024) | 2.70 | 2.99 | 3.21 | **2.97** | 0.603 | 0.552 | 91.3 |
| Gemini-2.5-flash (Comanici et al., 2025) | 2.81 | 3.29 | 3.25 | 2.88 | 0.600 | 0.561 | - |
| **Fine-tuned Model** | | | | | | | |
| HUMOR-CoT | 2.68 | **3.70** | 3.50 | 2.90 | **0.640** | 0.590 | 91.5 |
| CoT with Single Path (Kim et al., 2023) | 1.87 | 2.79 | 2.68 | 2.45 | 0.637 | 0.570 | 86.0 |
| CoT with Self-Improve (Chen et al., 2024a) | 2.38 | 3.68 | 3.00 | 2.65 | 0.629 | 0.578 | 89.1 |
| CoT with Subquestion (Wei et al., 2022) | 1.85 | 3.32 | 2.58 | 2.47 | 0.639 | **0.597** | 87.2 |
| HUMOR-RL (preview) | **2.83** | 3.67 | **3.55** | 2.79 | 0.631 | 0.588 | **92.3** |

**Results:** As summarized in Table 1, the proposed *HUMOR* framework achieves substantial improvements across multiple evaluation dimensions, validating its efficacy for humor-oriented meme generation. Specifically, in terms of *Humor*, *HUMOR-CoT* attains a score of 2.68, surpassing the base model Qwen2.5-7B-Instruct (2.39). Qualitative analysis suggests that *HUMOR*-improved models better capture nuanced humor mechanisms such as sarcasm and self-mockery. For **Readability**, *HUMOR-CoT* achieves a score of 3.70, outperforming all compared variants—including powerful closed-source models. It can generate meme texts with appropriate length and engaging structure, avoiding the verbosity common in many VLMs while maintaining humor expressivity, thereby better aligning with human writing conventions. It also excels in **theme relevance** and **originality**, demonstrating an ability to interpret deeper user intent rather than superficially referencing visual content. Although semantic similarity is less indicative for meme captions—which often consist of short phrases, *HUMOR-CoT* still achieves the closest alignment to reference captions among all models. Our proposed **Context-Swap Distance** metric further reveals that HUMOR-CoT (0.590) exceeds the baseline (0.564), indicating stronger generalization and context adaptability when user inputs are altered. This supports the hypothesis that hierarchical CoT reduces overfitting to concrete training labels. Finally, *HUMOR-CoT* achieves a human-likeness score of over 91%, significantly outperforming the base model (75.7%) and even surpassing the closed-source GPT-4o (91.3%).

Ablations on alternative CoT variants further illustrates the superiority of HUMOR: while *Single Path* lacks bottom-up visual grounding and produces narrow reasoning chains; *Self-Improve* attains high readability, it yields conservative, "safe but dull" outputs; *Subquestion* mitigates overfitting but suffers from over-decomposition, impairing humor and relevance. In contrast, HUMOR-CoT's two-stage reasoning more closely emulates human cognition process for better meme generation. Beyond human and text-level evaluations, we further validate model alignment through a VLM-based reclassification test (Appendix J.1). As summarized in Table 5, HUMOR-CoT consistently surpasses both the Qwen2.5-7B and Qwen2.5-32B base models across all four semantic dimensions—emotion, intention, theme, and style. Notably, despite being trained on the smaller 7B backbone, HUMOR-CoT even outperforms the 32B variant, demonstrating that the hierarchical CoT design contributes more effectively to user-intent preservation than scaling model size alone.

## 5.2 VLM Reliability Evaluation

After the CoT-based experiments, we further examined the reliability of VLM-based scoring for meme evaluation. In practice, existing VLMs often fail to align with human judgment: even for clearly distinct examples such as *In-the-wild Memes* (human-created and high-quality) versus *Text-Free Memes* (text removed), their absolute scores remain nearly identical, revealing that absolute scoring is inadequate for assessing humor or cultural nuance. As shown in Fig. 4(b), the group-wise relative ranking protocol produces much clearer distinctions between high- and low-quality memes and aligns well with human perception. A human study further validates that these rankings cap-

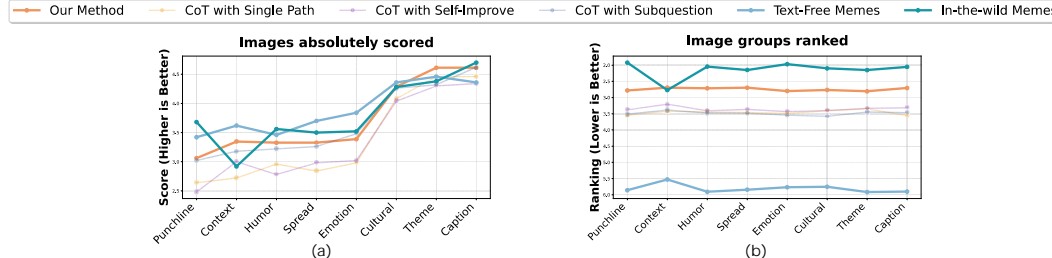

Figure 4: (a) VLM-based **absolute scoring** fails to distinguish meme quality. (b) **Group-wise ranking** produces more reliable distinctions, better aligned with human.

ture genuine preference structures, showing strong agreement with *Gemini-2.5-pro* (Spearman **0.72**, Kendall's $\tau$ **0.63**); full details are provided in Appendix L.4. Under this reliable evaluation protocol, HUMOR-CoT ranks second only to human-created memes and consistently surpasses all CoT-based training baselines. Building on this reliable ranking framework, we further assess HUMOR's ability to generalize to meme templates entirely unseen during training. We evaluate 20 novel templates with no image–text overlap with the training corpus. Gemini-2.5-pro jointly ranks outputs from different variants. As shown in Fig. 5, HUMOR-CoT again ranks second only to human-created memes, mirroring the in-distribution trend. This demonstrates strong zero-shot robustness: the hierarchical CoT effectively transfers its learned humor construction to unfamiliar formats rather than overfitting to template-specific patterns. For completeness, the full evaluation prompts are provided in Appendix M.4, detailed experimental settings in Appendix I.1, and representative outputs comparing different CoT reasoning schemes in Appendix K.1. Additional analyses—including risk-case identification (Appendix K.3), failure-case diagnostics (Appendix K.4), real-world application (Appendix K.5) and generalization to Unseen templates (Appendix K.2)—offer further qualitative and quantitative evidence supporting the robustness and interpretability of HUMOR-CoT.

### 5.3 REWARD MODEL RANK CONSISTENCY AND RL TRAINING

Table 2 evaluates reward models trained using the group-wise ranking strategy described above. These models are fine-tuned on different base models to align with human preference rankings.See Appendix H.2 for detailed training settings. For evaluation, we employ five meme templates: *Image1–Image5*. Each containing 10–15 candidate memes (see Figure 12). For every template, we obtain a *group-level* human ranking via MaxDiff (Appendix I.2).The human rankings for the example templates are shown in Appendix J.2. Model rankings are produced by: (i) collecting in-group pairwise comparison from either the base model or the fine-tuned reward model (*HUMOR-RM*), and (ii) aggregating them with Expected Borda Count (EBC) to acquire more reasonable sequence ranking. We report Kendall's $\tau$ and its $p$-value to test the rank consistency objective (Section 4.2). *HUMOR-RM* on *Keye-VL-8B* achieves consistently high $\tau$ with significant $p$-values (often $p \leq 10^{-3}$) across Image1–Image5, indicating strong within-group agreement with human preferences. On *Qwen2.5-VL-7B*, results are mixed-showing moderate alignment in some cases but near-chance level in others, with inconsistent significance. *Qwen2.5-VL-32B* and other backbones show limited gains. Overall, all fine-tuned models demonstrate improvements over their base versions under the same training and ranking supervision. However, the degree of rank consistency depends on the base model: semantically stronger and better-aligned backbones yield more reliable results, whereas weaker models align less steadily. We further validate the effectiveness of combining *HUMOR-RM* with a newly designed content reward (Appendix F) for RL training. Regarding content reward evaluation, see Appendix F.2 for the selection of evaluation models and the test of evaluation consistency. For the validity test of this part of content reward, please see Appendix H. As shown in Table 1, the resulting preview model exhibits enhanced performance in humor, relevance, and human rate.

### 5.4 REWARD MODEL ANALYSIS ON DIFFERENT BASE MODEL

Across all evaluated templates (Image 1–5), the *Keye-VL-8B* base model achieves higher in-group ranking consistency with human preferences than *Qwen2.5-VL* variants. We next examine why the post-training trajectories differ across base models and whether our training scheme induces model-specific preferences. Here, we present the differences among the top-ranked images preferred by reward models fine-tuned on different base models. As illustrated in Figure 6, *Qwen2.5-VL-7B* tends

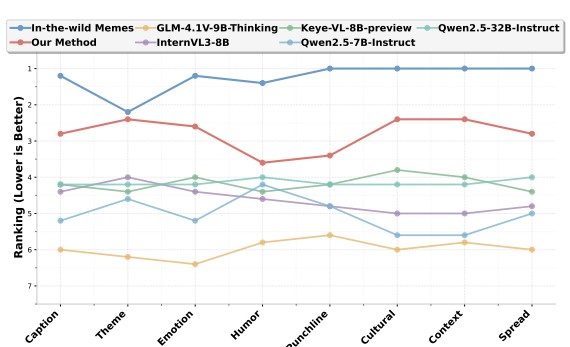
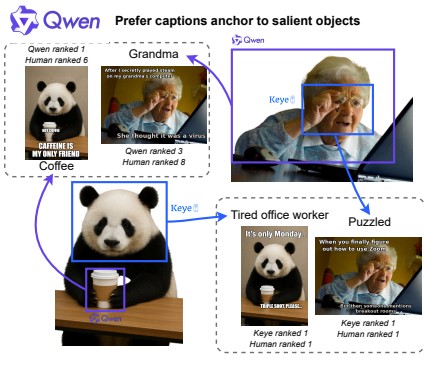

Figure 5: Group-wise ranking results on 20 unseen meme templates. Lower is better. HUMOR-CoT generalizes well and remains competitive with human-created memes.

Figure 6: *Qwen2.5-VL-7B* prefers captions that mention direct objects, whereas *Keye-VL-8B* prefers captions reflecting the human-like perception and understanding.

Table 2: (Reward Model) Ranking results of different baselines among distinct template images. It indicates the change after fine-tuning relative to the baseline: an increase in Kendall tau $\tau$ and a decrease in p-value $p$ represent improvements (highlighted in green), while the opposite indicates deterioration (shown in red). Significance levels: * $p < 0.05$; ** $p < 0.01$; *** $p < 0.001$.

| Model | Template 1 | | Template 2 | | Template 3 | | Template 4 | | Template 5 | |
|---|---|---|---|---|---|---|---|---|---|---|
| | $\tau \uparrow$ | $p \downarrow$ | $\tau \uparrow$ | $p \downarrow$ | $\tau \uparrow$ | $p \downarrow$ | $\tau \uparrow$ | $p \downarrow$ | $\tau \uparrow$ | $p \downarrow$ |
| **Qwen2.5-VL-7B (Base)** | 0.16 | 0.60 | 0.28 | 0.17 | 0.47 | 0.07 | -0.10 | 0.63 | 0.29 | 0.29 |
| **Qwen2.5-VL-7B (Finetuned)** | 0.47 | 0.07 | 0.56 | 0.03* | 0.42 | 0.11 | 0.14 | 0.50 | 0.47 | 0.07 |
| $\triangle$ vs Base | +0.31 | −0.53 | +0.28 | −0.14 | −0.04 | +0.04 | +0.25 | −0.13 | +0.18 | −0.22 |
| **Qwen2.5-VL-32B (Base)** | 0.16 | 0.61 | 0.16 | 0.44 | -0.02 | 1.00 | 0.14 | 0.50 | 0.29 | 0.29 |
| **Qwen2.5-VL-32B (Finetuned)** | 0.29 | 0.29 | 0.47 | 0.02* | 0.07 | 0.86 | 0.30 | 0.14 | 0.42 | 0.11 |
| $\triangle$ vs Base | +0.13 | −0.32 | +0.30 | −0.42 | +0.09 | −0.14 | +0.15 | −0.36 | +0.13 | −0.18 |
| **Keye-VL-8B (Base)** | 0.05 | 0.85 | 0.09 | 0.70 | 0.16 | 0.60 | 0.29 | 0.29 | 0.16 | 0.60 |
| **Keye-VL-8B (Finetuned)** | 0.78 | 0.00*** | 0.77 | 0.00*** | 0.78 | 0.00*** | 0.78 | 0.00*** | 0.78 | 0.00*** |
| $\triangle$ vs Base | +0.73 | −0.84 | +0.69 | −0.70 | +0.62 | −0.60 | +0.49 | −0.29 | +0.62 | −0.60 |

to anchor caption preferences on salient visual objects. For instance, when Image 5 depicts a panda holding a coffee cup, it favors captions containing the word "coffee"; Similarly, for Image 2, which shows an older woman looking at a laptop, it prefers references for "grandma" or computer-related terms. In contrast, *Keye-VL-8B* more consistently captures implied internal states or situational cues within the scene and aligns them with the template's communicative intent. In the same examples, it interprets the panda as resembling a "tired office worker" and the woman as appearing "puzzled", which aligns better with human rankings under our within-group evaluation protocol. These findings aligns with our theoretical expectation: while the reward model supplies only a preference ordering, effective alignment ultimately depends on the base model's capacity to represent the nuanced cues underlying human humor perception.

## 6 CONCLUSION

In this work, we tackled the complex challenge of teaching VLMs the art of in-the-wild meme generation, a task that requires nuanced reasoning beyond standard image captioning. Our proposed framework, *HUMOR*, successfully bridges the gap from visual perception to humorous punchline by instituting a two-stage process of hierarchical reasoning and preference alignment. Through a novel hierarchical CoT, the model learns to explore diverse creative paths while anchoring on high-quality outcomes. Furthermore, by leveraging group-wise preference modeling and RL, we ensure the generated humor aligns with human judgment in a stable and consistent manner. This work establishes a general and effective paradigm for open-ended multimodal generation tasks.

## LLM USAGE STATEMENT

We employ vision–language models (VLMs) for data preprocessing and evaluation. Specifically, we use *Doubao* to perform label assignment and generate hierarchical CoT traces for training data; at evaluation time, we use *Qwen-VL*, *Keye-VL*, and *Gemini-2.5-pro* as VLM judges to assess generated memes. For writing clarity only, we use *GPT-5* to polish the paper's wording without changing technical content or claims.

## ETHIC STATEMENT

All datasets used in this work are publicly available and licensed for research use. No private, personal, or biometric information is included. We adhered to all dataset terms of use and copyright requirements.

For the human evaluation study, all participants were recruited through legitimate platforms (e.g., Prolific/MTurk/University pool) and compensated at fair market rates. Before starting the survey, participants were provided with a clear informed-consent form explaining the study purpose, data usage, voluntary participation, withdrawal rights, and anonymity guarantees. No personally identifying information was collected, and all responses were fully anonymized.

During data preprocessing, we removed violent, hateful, and other harmful content to the best extent possible. Because meme-generation systems may still produce biased or sensitive content, we acknowledge potential risks related to discrimination or fairness. To mitigate these risks, we recommend standard safety measures (automated content filters, human-in-the-loop review, and clear usage policies) when deploying the model.

Our research complies with ethical guidelines for human-subject research and responsible AI development. No data will be released that could enable misuse.

## REPRODUCIBILITY STATEMENT

Upon acceptance, we will release: (i) the full list of dataset sources we use; (ii) our constructed CoT supervision data and the pairwise/reward datasets; and (iii) the complete training and inference codebase. We will also provide prompts, hyperparameters, random seeds, model checkpoints (or scripts to reproduce them), and evaluation scripts to enable end-to-end replication.

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

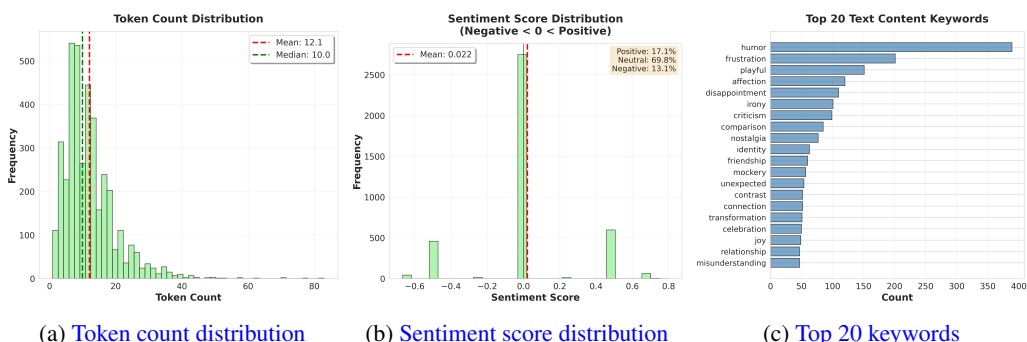

(a) Token count distribution  (b) Sentiment score distribution  (c) Top 20 keywords

Figure 7: Textual properties of meme captions in training dataset

# A  DATASET STATISTICS AND ANALYSIS

In this section, we provide a comprehensive analysis covering linguistic features, semantic content, and semantic diversity. These statistics validate that the dataset captures the nuanced, punchy, and diverse nature of internet humor required for training robust VLMs.

## A.1  LINGUISTIC AND SEMANTIC COMPOSITION

We first analyze the textual properties of the meme captions to ensure they align with the linguistic conventions of internet culture.

**Token Count Distribution:** As illustrated in Figure 7a, the token count follows a log-normal distribution with a mean of 12.1 and a median of 10.0. This confirms that the dataset consists predominantly of concise, high-impact text, consistent with the "short and punchy" nature of memes.

**Sentiment Distribution:** The sentiment analysis (Figure 7b) reveals a dominant Neutral class (69.8%), with balanced Positive (17.1%) and Negative (13.1%) tails. This heavy skew toward neutrality is expected and desirable; meme humor often relies on *deadpan* delivery or irony, where the text itself appears objective or factual, and the humor emerges only through the juxtaposition with visual context.

**Semantic Keywords:** The top-30 keyword analysis (Figure 7c) confirms that the dataset is grounded in abstract emotional concepts rather than merely descriptive tags. Dominant keywords such as *Humor*, *Frustration*, *Irony*, and *Disappointment* indicate that the data captures the core thematic essence of relatable internet memes.

## A.2  SEMANTIC DIVERSITY AND RATIONALITY OF DISTANCE

A critical quality of a high-quality meme dataset is **paraphrastic diversity**—the ability to express the same underlying template intent through varied textual realizations. To quantify this, we analyzed the distribution of semantic distances ($1 -$ Cosine Similarity) between captions within the dataset.

As shown in Figure 8, the distance metric follows a normal distribution with the following characteristics:

- **Central Tendency:** Both the mean and median are exactly **0.570**, with a standard deviation of **0.067**.
- **The "Goldilocks" Interval** $[0.5, 0.6]$**:** A significant majority of the data (52.5%) falls within this specific range.

**Rationality of the** $[0.5, 0.6]$ **Range:** We argue that this distance distribution is not only reasonable but indicative of a high-quality dataset for open-ended generation:

1. **Avoidance of Mode Collapse (**$> 0.1$**):** A very low distance (e.g., $< 0.2$) would imply that the dataset contains largely duplicate or repetitive captions, which leads to overfitting and lack

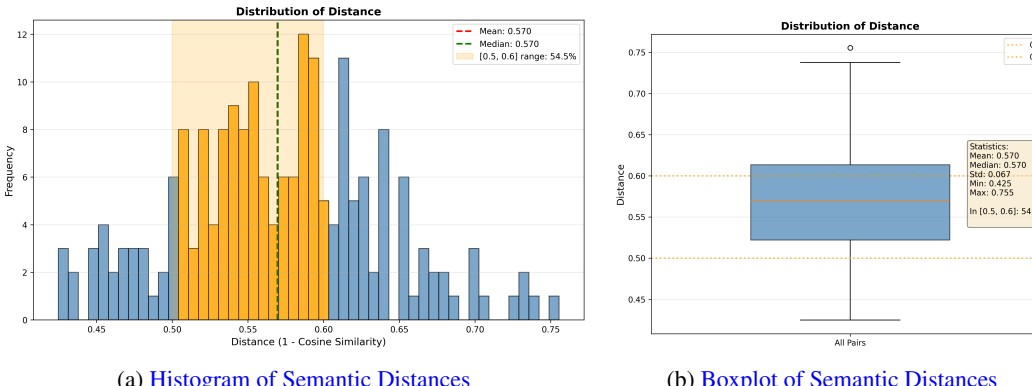

(a) Histogram of Semantic Distances    (b) Boxplot of Semantic Distances

Figure 8: **Analysis of Semantic Diversity.** The distribution of semantic distances (defined as $1 -$ Cosine Similarity) exhibits a mean and median of 0.570. The concentration of data ($54.5\%$) within the $[0.5, 0.6]$ interval indicates a healthy balance: the captions are semantically related enough to share a theme, yet diverse enough to avoid trivial repetition.

of creativity. Our distribution shows virtually no mass in this region, confirming high lexical diversity.

2. **Semantic Coherence** ($< 0.9$)**:** A very high distance (e.g., $> 0.8$) would suggest random or unrelated text. The maximum distance observed is 0.755, with the vast majority below 0.7, ensuring that the captions remain thematically grounded to the meme templates.

3. **Optimal Paraphrasing:** The concentration at $0.57$ represents an optimal middle ground where captions share the same latent humor or intent (lowering distance) but utilize distinct vocabulary and sentence structures (increasing distance). This supports our claim that the dataset facilitates learning robust, generalized humor representations rather than rote memorization.

## B    HIERARCHICAL CHAIN-OF-THOUGHTS OF METAPHOR

To enhance our model's understanding of humor, we replicated the human meme creation process. Through extensive analysis of human meme creation, we extracted a paradigm for hierarchical meme feature analysis.

Take the "Distracted Boyfriend" meme as an example. Humans first capture: the delighted expression of the woman on the left, the action of the man in the center looking back and his subtle flirtatious gaze, the annoyed posture of the woman on the right, and the triangular compositional relationship and explicit emotional direction formed by the three individuals. Humans further abstract this scene and discover that it can be applied to any scenario of infatuation with something new and abandonment of the old, establishing entity mapping relationships. Thus, when the user's request is workplace culture, this template can be adapted to depict a leader being attracted by a new employee during a meeting, with a senior employee showing an expression of helplessness, vividly illustrating the workplace "new vs. old" relationship and generating humor.

How would humans fill in the text? Through statistical analysis of 5,000 classic memes, we found that the text positions in common meme templates are fixed, and the text content is highly correlated with its position. For instance, in the "Distracted Boyfriend" template, the position corresponding to the woman on the right is often used to represent the neglected object, the position corresponding to the man in the center represents the subject of attention shift, and the position corresponding to the woman on the left is the newly focused entity. Therefore, we integrate "text content generation" and "text position allocation" in the meme generation process. By annotating text box positions in the image, the model only needs to use its inherent visual localization ability to find the boxes, understand that text needs to be written in specific areas, and then combine spatial semantic mapping relationships to generate text with greater humorous effects in these positions.

We aim to imitate this thought process to construct Chain-of-Thought (CoT) data:

**Data Collection and Preprocessing**

**Meme Images** We collected over 4,000 meme images from public dataset (Xu et al., 2024), and established a multi-dimensional labeling system:

1. **Emotion Classification**: Covers 7 basic emotions and intensity levels.
2. **Intent Detection**: Differentiates between 10 creation intents such as offense and entertainment.
3. **Metaphor Analysis**: Records metaphorical entities and cross-domain mapping relationships.

**Safety-Driven Dataset Cleaning.** To mitigate potential risks within the raw dataset—such as political bias, sexually explicit content, and sensitive themes like discrimination—we implemented an automated filtering protocol leveraging the intrinsic safety guardrails of the VLM API (e.g. `doubao-1.5-vision-pro`). Specifically, during the image understanding phase, we prompted the API to interpret each meme. We adopted a "refusal-based" criterion: instances where the API triggered a safety warning or refused to generate a response were flagged as containing harmful or negative content. These samples were systematically excluded from our training corpus to ensure compliance with ethical safety standards.

**Base Images and Text Content/Position Information** The FLUX.1-dev-Controlnet-Inpainting-Beta model is used to erase and restore the text areas in original memes, obtaining text-free base images. Meanwhile, OCR technology precisely records the (position, content) pairs of text, providing spatial semantic data for subsequent training.

**User Requirements** We reconstructed user requirements in reverse using APIs. Taking the meme's labels and final text as inputs, we utilized prompts to reverse-engineer the user's initial request. We analyzed the following dimensions of user requirements: emotion category, emotion intensity, intention, Scene or theme , style preference, and keywords.

**CoT Data Generation**

**Stage One** Using the base image as input, we extract high-level semantics of the meme.

First, we perform visual element decomposition. Our framework systematically deconstructs meme templates from four key visual dimensions:

1. **Main Subject Characteristics**: Analyze facial expressions, poses, clothing, and dynamic relationships between characters.
2. **Composition Logic**: Identify visual focal points, color contrasts, and spatial relationships.
3. **Cultural Markers**: Recognize identifiable meme formats and pop culture references.
4. **Narrative Threads**: Interpret body language implications and prop symbolism.

Then, we conduct scenario association and humor construction based on visual analysis:

1. **Social Contexts**: Identify scenarios suitable for group chats, comment sections, and private conversations.
2. **Topic Relevance**: Establish connections with workplace culture, life dilemmas, and internet hotspots.
3. **Emotional Mapping**: Determine appropriate humor techniques, including satire, self-deprecation, exaggeration, and contrast.

**Stage Two** Using the base image analysis from Stage One, user requirements, and final text as inputs, we infer the customized creation process for specific requests.

We provide few-shot examples of this parsing process. For instance, for the "Distracted Boyfriend" meme, when Stage One yields the semantic pattern of infatuation with something new and abandonment of the old, and identifies three entity positions: A [attention-shifting subject], B [newly focused entity], and C [neglected object], the user's request is a technology theme with the keyword "Apple fanatic." We consider how to align the expression of infatuation with something new and abandonment of the old with the context of technology product updates to reflect being an Apple fanatic. We

infer that the semantic mapping of new and old phones is similar. Therefore, combining this image, we deduce that the text should be filled as: "A: APPLE FANS, B: IPHONE 11, C: IPHONE 10," humorously expressing enthusiasm for Apple's new technological products.

**Training Rationale and Process**  We conduct instruction-tuning training using CoT data as supervisory signals. Since our training data contains numerous instances of the same base image, the two-stage CoT process essentially learns metaphorical semantic relationships across different scenarios. It is a divergent associative thinking training where one base image corresponds to multiple scenarios. This CoT approach not only enables the model to understand the high-level semantics of the image itself but also establishes multi-scenario associative capabilities.

**Determination and Extraction of Generated Text Format**  Text boxes in the image are marked using a top-to-bottom, left-to-right coordinate sorting rule, and text content is recorded in the labels in order and in box format. The prompt explicitly requires the model to output in the format "box1:text1, box2:text2."

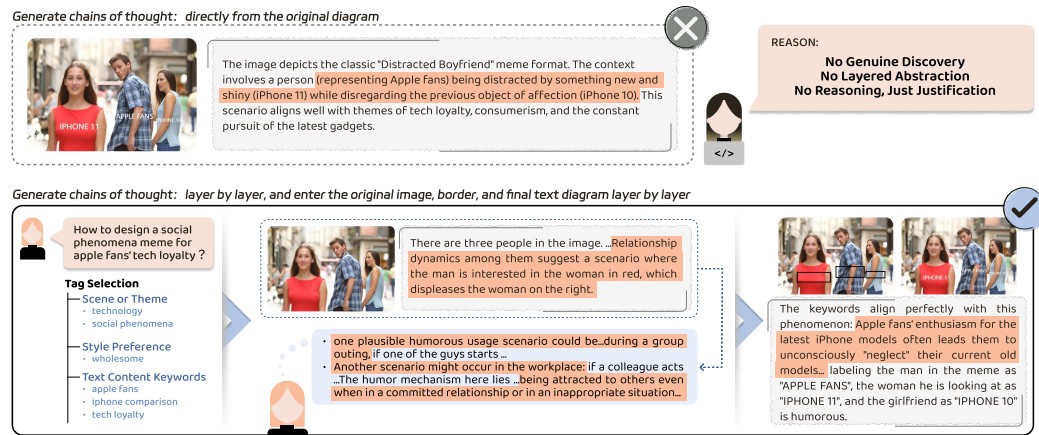

Figure 9: Comparison between direct CoT generation from the original image and our hierarchical CoT generation approach.

**Critical Comparison: Direct vs. Hierarchical CoT**  The direct approach of generating chains of thought from the original image is essentially reverse engineering rather than genuine reasoning. It suffers from four critical flaws: 1) **No Genuine Discovery**: it skips the exploratory stage where humor emerges from active associative search, jumping straight to a fixed answer; 2) **No Layered Abstraction**: it leaps from raw visual details to a specific conclusion without building transferable intermediate metaphors; 3) **No Reasoning, Just Justification**: instead of true inference, it merely defends a predetermined conclusion.

In contrast, our layered CoT framework mirrors human reasoning by progressively abstracting from visual description to general metaphorical patterns and then to domain-specific humor instantiations, thereby enabling genuine creativity and robust generalization.

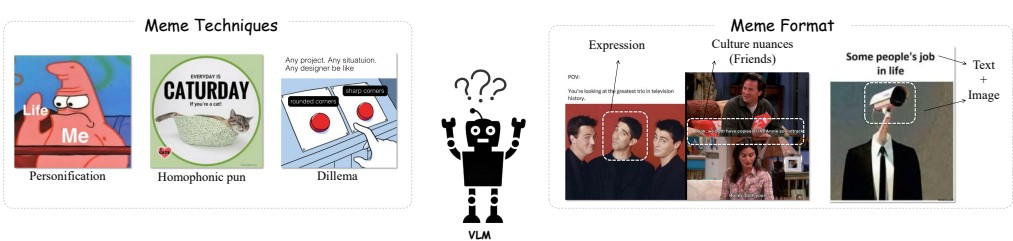

Figure 10: Examples of memes common on the internet

# C    REWARD MODELING: ASSUMPTIONS AND PROOFS

## C.1    SETUP AND ASSUMPTIONS

For a fixed group $G$, the latent humor functional is $h_G : G \to [0,1]$. Pairwise labels follow the observation model of Eq. (1):

$$\Pr[m_i \succ m_j \mid G] = \Lambda\big(h_G(m_i) - h_G(m_j)\big),$$

where $\Lambda : \mathbb{R} \to (0,1)$ is strictly increasing. A reward model maps a meme $m = (I, c)$ to a score $s_\phi(m)$; the pairwise probability is

$$\hat{p}_{ij}^G = \sigma\big(s_\phi(m_i) - s_\phi(m_j)\big),$$

and $\phi$ is learned by minimizing the empirical pairwise cross-entropy $\mathcal{L}_{\text{pair}}$. We assume (A1) the data contains i.i.d. pairs drawn within $G$ with non-degenerate coverage; (A2) the model class for $s_\phi$ is rich enough to fit the Bayes-optimal decision boundary; (A3) identifiability is up to an additive constant per group (sufficient for ranking).

## C.2    RANK CONSISTENCY (PROPOSITION 1) — PROOF

*Proposition* (Rank consistency (main text Proposition 1)). Under Eq. (1) with strictly increasing $\Lambda$, any risk minimizer of the logistic pairwise loss recovers the same within-group ordering as $h_G$.

**Proof.**    Let $\eta_{ij} = \Pr[m_i \succ m_j \mid G] = \Lambda(\Delta_{ij})$ with $\Delta_{ij} = h_G(m_i) - h_G(m_j)$. The Bayes-optimal pairwise classifier for logistic loss satisfies $\sigma(s_i^\star - s_j^\star) = \eta_{ij}$, hence

$$s_i^\star - s_j^\star = \sigma^{-1}(\eta_{ij}) = \sigma^{-1}\big(\Lambda(\Delta_{ij})\big) =: \psi(\Delta_{ij}),$$

where $\psi$ is strictly increasing as a composition of strictly increasing functions. Therefore

$$s_i^\star - s_j^\star > 0 \iff \Delta_{ij} > 0 \iff h_G(m_i) > h_G(m_j).$$

Thus any minimizer (up to additive constants) induces the same strict order as $h_G$ inside $G$.    □

## C.3    NOISE ROBUSTNESS (PROPOSITION 2) — PROOF

*Proposition* (Noise robustness (main text Proposition 2)). Let $\Delta_{ij}^G = |h_G(m_i) - h_G(m_j)|$. Suppose the learned classifier has average pairwise error $\varepsilon$. If we split pairs into "small-margin" ($\Delta_{ij}^G < \delta$) and "large-margin" ($\Delta_{ij}^G \geq \delta$), then the reversal probability obeys

$$\Pr[\text{reversal}] \leq \Pr[\Delta_{ij}^G < \delta] + \Pr[\text{reversal} \mid \Delta_{ij}^G \geq \delta] \leq \Pr[\Delta_{ij}^G < \delta] + \varepsilon_\delta,$$

where $\varepsilon_\delta$ decreases as $\delta$ increases and increases with the classifier error $\varepsilon$; in particular, under the observation model Eq. (1), the conditional flipping probability on large-margin pairs is upper-bounded by a monotonically decreasing function of $\delta$.

**Proof.**    Let $K$ be the event "classifier reverses the true order". Decompose by a margin threshold $\delta > 0$:

$$\Pr[K] = \Pr[K \wedge (\Delta_{ij}^G < \delta)] + \Pr[K \wedge (\Delta_{ij}^G \geq \delta)] \leq \Pr[\Delta_{ij}^G < \delta] + \Pr[K \mid \Delta_{ij}^G \geq \delta].$$

The second term is at most the classifier's conditional error on large-margin pairs, denoted $\varepsilon_\delta$. Under Eq. (1), the Bayes error on a pair decreases monotonically with $|\Delta_{ij}^G|$, hence $\varepsilon_\delta$ decreases in $\delta$. If the global average error is $\varepsilon$, then $\varepsilon_\delta \leq \varepsilon$ and often much smaller. Thus large true gaps are stably preserved, while flips concentrate on small-margin pairs.    □

## C.4    FROM PAIRWISE TO GROUP RANKING (EBC)

Given sparsity, we aggregate pairwise probabilities into a within-group ranking via Expected Borda Count (EBC): each item's score equals its expected number of wins against others according to $\hat{p}_{ij}^G$. EBC is a monotone transformation of the empirical pairwise preferences and inherits rank consistency in expectation when the pairwise model is consistent, providing a coherent group-wise order for evaluation and optimization. (Operational details as in Sec. 4.2.)

## D  GROUP-WISE POLICY OPTIMIZATION (GRPO): GUARANTEES AND PROOFS

### D.1  OBJECTIVE AND NOTATION

For a candidate set $\mathcal{S}_G$ with group ranking distribution $q_G$ (from EBC), the GRPO loss is

$$\mathcal{L}_{\text{GRPO}}(\theta) = \mathbb{E}_{(I,G)}\Big[ - \sum_{m_k \in \mathcal{S}_G} q_G(m_k) \log \pi_\theta(c_k \mid I) \Big] + \beta\, \mathbb{E}_I\big[ \text{KL}\big(\pi_\theta(\cdot \mid I) \,\|\, \pi_{\text{ref}}(\cdot \mid I)\big)\big].$$

Intuitively, the first term pushes $\pi_\theta$ toward $q_G$ within the group (listwise), and the KL term limits drift from a safe reference policy $\pi_{\text{ref}}$; both are group-local, matching comparability in our formulation (Sec. 3).

### D.2  BOUNDED DEGRADATION VIA KL CONTROL

We formalize the "cannot degrade beyond a bounded amount" claim under bounded KL.

*Proposition* (Bounded improvement under GRPO (main text Proposition 2)). Assume the reward model is rank-consistent (Proposition C.2) and $h_G \in [0,1]$. Let $\Delta_{\text{KL}} = \mathbb{E}_I\big[\text{KL}\big(\pi_\theta(\cdot \mid I) \,\|\, \pi_{\text{ref}}(\cdot \mid I)\big)\big]$. Then the expected within-group humor satisfies

$$\mathbb{E}_{(I,G)}\Big[\mathbb{E}_{c \sim \pi_\theta(\cdot \mid I)}\, h_G\big((I,c)\big)\Big] \ \geq\ \mathbb{E}_{(I,G)}\Big[\mathbb{E}_{c \sim \pi_{\text{ref}}(\cdot \mid I)}\, h_G\big((I,c)\big)\Big]\ -\ \sqrt{\tfrac{1}{2}\,\Delta_{\text{KL}}}.$$

Consequently, if GRPO enforces $\Delta_{\text{KL}} \leq \tau$ (by choosing $\beta$ or an explicit trust region), the expected humor cannot drop by more than $\sqrt{\tau/2}$; with rank-consistent $q_G$, optimization increases the probability of higher-$h_G$ captions, so the net effect is non-decreasing or improved expected humor once the pull toward $q_G$ outweighs this bound.

**Proof.**  For any fixed $(I,G)$, Pinsker's inequality gives

$$\big\|\pi_\theta(\cdot \mid I) - \pi_{\text{ref}}(\cdot \mid I)\big\|_{\text{TV}} \ \leq\ \sqrt{\tfrac{1}{2}\,\text{KL}\big(\pi_\theta(\cdot \mid I) \,\|\, \pi_{\text{ref}}(\cdot \mid I)\big)}.$$

Since $h_G \in [0,1]$, by the variational characterization of total variation for bounded functions,

$$\Big|\mathbb{E}_{\pi_\theta}[h_G] - \mathbb{E}_{\pi_{\text{ref}}}[h_G]\Big| \ \leq\ \big\|\pi_\theta - \pi_{\text{ref}}\big\|_{\text{TV}} \ \leq\ \sqrt{\tfrac{1}{2}\,\text{KL}(\pi_\theta \| \pi_{\text{ref}})}.$$

Averaging over $(I,G)$ yields the stated bound. During GRPO, the cross-entropy term $-\sum q_G \log \pi_\theta$ (with rank-consistent $q_G$) increases mass on higher-$h_G$ captions within the group, while the KL term keeps the deviation controlled. Thus expected humor cannot deteriorate beyond the Pinsker bound and, in practice, improves as the listwise alignment progresses. $\square$

### D.3  DISCUSSION: WHY LISTWISE $q_G$ MATTERS

Because $q_G$ aggregates pairwise signals into a coherent group distribution consistent with $h_G$'s ordering, the CE term directly performs a proximal step toward the better subset of captions *without* inventing any cross-group scale. This matches our problem scope and the guarantees in Sec. 4.2–4.3 of the main text.

## E  PAIR-WISE DATASET CONSTRUCTION

Our reward model is trained on *pairwise* comparisons. Intuitively, pairs whose ordering is both *reliably correct* and *increasingly challenging* drive the model toward more consistent ranks. We therefore construct a curriculum of **five difficulty tiers**, guaranteeing correct orderings while progressively raising difficulty (from trivial mismatches to near-ties within the same template/scene). To span both trivial and subtle distinctions, we sample pairs across all tiers and upweight harder tiers during training, yielding a supervision signal that is confident yet discriminative:

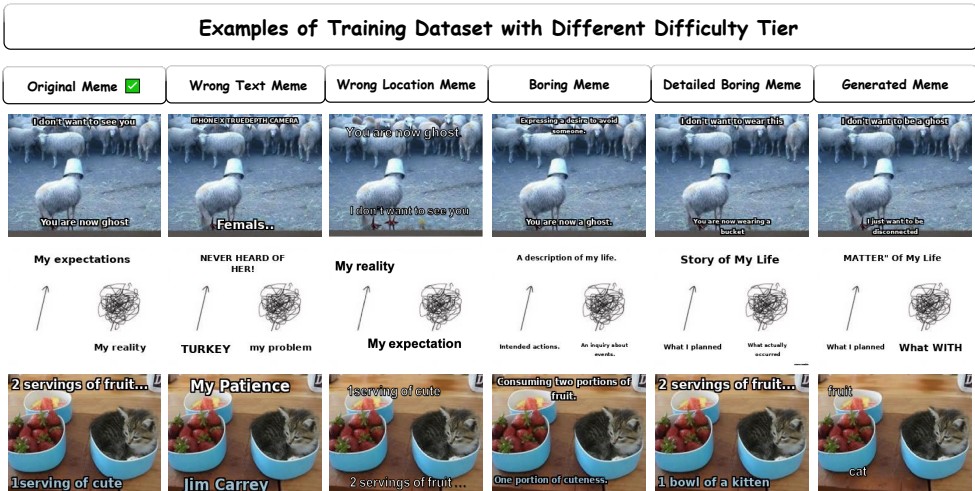

Figure 11: Examples of training datasets with different difficulty tier

1. Wrong Text Meme ($\star$): This is the most straightforward case, where the original text is replaced with unrelated content, completely removing the humor. This type of meme is easy for the model to classify as "non-humorous" and acts as a baseline.
2. Wrong Location Meme ($\star\star$): A slightly more complex case involves shifting the position of the text in the image. While the metaphor may still exist, the humor diminishes due to the misplacement of text. The model must learn that small positional changes can significantly impact the meme's humor, reflecting a higher degree of difficulty.
3. Boring Meme ($\star\star$): Here, the meme is altered to include a more mundane or less engaging version of the original text. This teaches the model to distinguish between "humorous" and "boring" versions of the same meme. Although the content still aligns with the original, the humor is less impactful, presenting a challenge for classification.
4. Detailed Boring Meme ($\star\star\star$): This is a more nuanced case where only one or two words are changed to make the meme less funny. Despite the minimal changes, the meme's humor is significantly affected. The classifier must be able to identify these subtle shifts in humor, marking this as a more difficult classification task.
5. Generated Meme ($\star \sim \star\star\star$): Finally, memes generated by the fine-tuned VLM represent the highest difficulty level. These memes are intended to be as humorous as the original meme, requiring the classifier to discern fine-grained differences in humor between the generated meme and the original. This provides the model with an opportunity to improve its sensitivity to subtle differences in meme quality.

The example of the training dataset is shown in Figure 11. By constructing a dataset with pairs of memes across these varying levels of humor, we enable the classifier to learn not only to distinguish obviously bad memes from good ones but also to understand the nuanced differences that make one meme more humorous than another. This rich dataset plays a crucial role in refining the reward model, allowing it to classify memes based on subtle human preferences.

We stratify training so each mini-batch contains an equal number from each tier.

## F AUXILIARY REWARDS FOR REASONING-PATH OPTIMIZATION

While optimizing toward the group-wise reward induced by the reward model (Sec. 4.2) is theoretically sufficient to improve the quality of generated memes, the reinforcement learning stage does not directly supervise the internal reasoning path $r = (r_{\text{tmpl}}, r_{\text{scene}})$ because the primary feedback is attached to the realized meme $(I, c)$. To explicitly shape the quality of the reasoning process itself, we introduce two auxiliary rewards that operate on $r$: a *format reward* and a *content reward*.

| Source Model | Target Score | Qwen-VL | Keye-VL | Logits (Qwen) |
|---|---|---|---|---|
| Qwen-VL | 0 | 0.692 | 0.209 | 0.498 |
| | 0.2 | 0.642 | 0.229 | 0.452 |
| | 0.6 | 0.678 | 0.096 | 0.383 |
| | 0.8 | 0.522 | 0.400 | 0.368 |
| | 1 | 0.630 | 0.478 | 0.387 |
| Keye-VL | 0 | 0.508 | 0.280 | 0.390 |
| | 0.2 | 0.695 | 0.653 | 0.379 |
| | 0.6 | 0.672 | 0.731 | 0.412 |
| | 0.8 | 0.674 | 0.769 | 0.388 |
| | 1 | 0.538 | 0.691 | 0.462 |

Table 3: Content reward evaluation across different target quality levels. Keye-VL as judge exhibits the clearest monotonic trend, and is therefore adopted as our content reward model in RL.

### F.1 FORMAT REWARD

The format reward enforces structural completeness of the CoT to ensure that essential modules appear and are well-formed. It is computed by deterministic string/structure matching without using LLM-as-judge. Concretely, given a sampled reasoning trace $r$ for $(I, U)$, we check:

1. **Presence of mandatory sections** (e.g., a `Comprehensive Description` section that summarizes visual content and intended template-level intent).
2. **Two-stage structure** (explicit evidence of both template-level intent and context-level grounding consistent with Sec. 4.1).
3. **Text-on-Meme box formatting** (the `Text on the Meme` block must specify box–text mappings consistent with the bounding boxes $B = \{b_i\}$ so that rendered text $T = \{t_i\}$ aligns with $B$).

The format reward $R_{\text{fmt}}(r) \in [0, 1]$ is the normalized sum of satisfied checks. It shapes $r$ toward complete and renderable reasoning without requiring any subjective judgment.

### F.2 CONTENT REWARD

The content reward evaluates the informativeness and plausibility of the CoT content via an *LLM-as-judge*. We prompt an evaluation model to score $r$ along four interpretable dimensions (e.g., visual grounding, template intent clarity, metaphorical mapping, and punchline coherence), each with discrete bands (e.g., 1/4/7 points with band descriptors such as "no object description / coarse description / detailed object attributes"). Scores are summed and rescaled to $R_{\text{cnt}}(r) \in [0, 1]$.

However, prior work rarely verifies whether a vision–language reward signal is monotonic with respect to intended semantic quality. To ensure that our RL optimization is grounded on a reliable content metric, we systematically compare several candidate reward options.

We construct five groups of captions whose intended content quality is controlled at target levels $\{0, 10, 30, 40, 50\}$. These groups are obtained by prompting two widely used multimodal LLMs—*Qwen2.5-VL-7B* and *Keye-VL-8B*—to generate CoT rationales and captions under progressively stronger quality constraints. For each generated caption, we compute content reward using three scoring strategies: *Qwen2.5-VL-7B* scorer, *Keye-VL-8B* scorer, and *Qwen2.5-VL-7B* output logits, where the final score is calculated from the normalization of logits of each score token.

In Table 3, we then examine whether the final reward values increase along with the intended quality levels. Across both data sources (Qwen-generated and Keye-generated), *Keye-VL-8B* as the judge exhibits the clearest monotonic trend: scores grow consistently as target quality increases. In contrast, *Qwen2.5-VL-7B* scoring shows weaker correlation, and normalized logits are noticeably noisy. Notably, *Keye-VL-8B* remains stable even when scoring content generated by another model, suggesting better cross-distribution generalization.

These results indicate that *Keye-VL-8B* provides the most rank-consistent, semantically aligned content reward, and we therefore adopt it as the content reward model in our RL stage.

## F.3 Integration with GRPO

Let $s_{\mathrm{RM}}(m)$ denote the reward-model score that induces the group-wise ranking distribution $q_G$ via EBC in Sec. 4.2. For a candidate set $\mathcal{S}_G = \{m_k = (I, c_k)\}$ with associated reasoning traces $\{r_k\}$, we construct an *augmented* group-wise target $\tilde{q}_G$ by combining the primary signal with auxiliary rewards on $r_k$:

$$\tilde{q}_G(m_k) \;\propto\; \exp\Big(\tfrac{1}{\tau}\Big[s_{\mathrm{RM}}(m_k) \;+\; \lambda_{\mathrm{fmt}}R_{\mathrm{fmt}}(r_k) \;+\; \lambda_{\mathrm{cnt}}R_{\mathrm{cnt}}(r_k)\Big]\Big), \qquad \sum_{m_k \in \mathcal{S}_G} \tilde{q}_G(m_k) = 1, \tag{5}$$

where $\tau > 0$ is a temperature and $\lambda_{\mathrm{fmt}}, \lambda_{\mathrm{cnt}} \geq 0$ are weights. The GRPO objective in Eq. equation 4 is then used with $q_G$ replaced by $\tilde{q}_G$.

*Remark* (Isotonic shaping and theoretical guarantees). If $(\lambda_{\mathrm{fmt}}, \lambda_{\mathrm{cnt}})$ are chosen such that Eq. 5 is an *isotonic* transformation of the reward-model ranking (i.e., it does not invert the order implied by $s_{\mathrm{RM}}$ except to break ties among near-equal items), then the rank consistency guarantees stemming from Proposition 2 are preserved in expectation. Moreover, the KL-bounded improvement in Proposition 4 continues to hold because the proof relies on boundedness of $h_G$ and a KL constraint, both unaffected by auxiliary shaping. In practice we set $\lambda_{\mathrm{fmt}}, \lambda_{\mathrm{cnt}}$ small and use them primarily as tie-breakers and regularizers over $r$, which empirically reduces variance and accelerates convergence without altering the main ordering.

## G EBC Aggregation

**Definition (Expected Borda Count).** Given a group $G$ and a finite candidate set $\mathcal{S}_G = \{m_1, \ldots, m_n\}$ with pairwise preference probabilities $\widehat{p}_{ij}^G = \Pr[m_i \succ m_j]$, the Expected Borda Count of item $m_i$ is

$$\mathrm{EBC}_G(m_i) \;=\; \sum_{\substack{j=1 \\ j \neq i}}^{n} \widehat{p}_{ij}^G.$$

Ties or missing edges are handled by omitting terms (equivalently, treating $\widehat{p}_{ij}^G$ as undefined); in evaluation we normalize by the number of available opponents for $m_i$.

**Basic properties.** (i) If all $\widehat{p}_{ij}^G \in \{0, 1\}$, EBC reduces to the classical Borda score (number of wins). (ii) If there exists a latent utility $u : \mathcal{S}_G \to \mathbb{R}$ such that $\widehat{p}_{ij}^G = \sigma(u(m_i) - u(m_j))$ with strictly increasing $\sigma$, then sorting by EBC is order-equivalent to sorting by $\sum_{j \neq i} \sigma(u(m_i) - u(m_j))$; in particular, when gaps are consistent across pairs, the EBC order agrees with the order of $u$. (iii) Under independent edge noise and bounded missingness, the variance of $\mathrm{EBC}_G(m_i)$ decreases with the number of observed pairs, making the aggregate rank more stable than any single comparison.

**Listwise normalization (optional).** For downstream use, one may define a soft distribution over $\mathcal{S}_G$ via a temperature $T > 0$:

$$q_G(m_i) \;=\; \frac{\exp\big(\mathrm{EBC}_G(m_i)/T\big)}{\sum_{k=1}^{n} \exp\big(\mathrm{EBC}_G(m_k)/T\big)},$$

which converts EBC scores into smooth listwise targets for within-group reweighting. This preserves the group-local nature of the signal and avoids inventing cross-group scales.

**Notes on implementation.** We compute $\widehat{p}_{ij}^G$ only within groups and on the (usually small) candidate sets used for evaluation or optimization. When the pair graph is sparse, we keep EBC unbiased by summing over observed opponents and normalizing by their count; when required, we add small-degree regularization to avoid over-confident ranks for items with very few edges. The pseudocode is shown in the Algorithm 1

---

**Algorithm 1** Expected Borda Count (matrix form)

---

**Require:** Candidate set $\mathcal{S}_G = \{m_1, \ldots, m_n\}$; pairwise estimates $\widehat{p}_{ij}^G = \Pr[m_i \succ m_j]$ (may be undefined); temperature $T > 0$ (optional); small-degree regularizer $\alpha \geq 0$ (optional).
**Ensure:** EBC scores $\mathrm{EBC}_G(m_i)$ for all $m_i$; optionally soft listwise $q_G(m_i)$.
1: Initialize $EBC[i] \leftarrow 0$ and $deg[i] \leftarrow 0$ for all $i \in \{1, \ldots, n\}$.
2: **for** $i = 1$ to $n$ **do**
3:     **for** $j = 1$ to $n$ **do**
4:         **if** $i = j$ **then**
5:             **continue**
6:         **end if**
7:         **if** $\widehat{p}_{ij}^G$ is defined **then**           $\triangleright$ omit ties/missing edges
8:             $EBC[i] \leftarrow EBC[i] + \widehat{p}_{ij}^G$
9:             $deg[i] \leftarrow deg[i] + 1$
10:        **end if**
11:     **end for**
12: **end for**
13: **for** $i = 1$ to $n$ **do**           $\triangleright$ unbiased normalization under sparsity
14:     **if** $deg[i] > 0$ **then**
15:         $EBC[i] \leftarrow \dfrac{EBC[i] + \alpha}{deg[i] + \alpha}$      $\triangleright$ $\alpha$ prevents overconfidence at tiny degree
16:     **else**
17:         $EBC[i] \leftarrow 0$
18:     **end if**
19: **end for**
20: **if** $T$ is provided **then**
21:     compute $q[i] \leftarrow \exp(EBC[i]/T)$ for all $i$, then $Z \leftarrow \sum_k q[k]$
22:     **return** $(EBC[i], q[i] \leftarrow q[i]/Z)$ for all $i$
23: **else**
24:     **return** $EBC[i]$ for all $i$
25: **end if**

---

# H   TRAINING SETTINGS

## H.1   COT SUPERVISED FINE-TUNING SETTINGS

The experimental settings for cot supervision and fine-tuning are shown in Table 4

## H.2   REWARD MODEL TRAINING SETTINGS

Our reward model is implemented as a lightweight extension on top of the base vision–language models. Concretely, we take the final hidden embedding of the last transformer layer and append a two-way classification head. This simple design allows the model to learn preference signals while reusing the representational power of the pretrained backbone.

Based on the dataset constructed in Appendix E, we train reward models using the *LLaMA-Factory* framework with the following backbones: *Keye-VL*, *Qwen2.5-VL-7B*, and *Qwen2.5-VL-32B*. All models are fine-tuned with LoRA ($r = 8$, lora target is all) to reduce memory and computation overhead. We adopt a learning rate of $1 \times 10^{-4}$, with a warmup ratio of 0.1. Each model is trained on a single NVIDIA A800 GPU.

# I   EVALUATION SETTINGS

## I.1   VLM EVALUATES EXPERIMENTAL SETUP

**Evaluation Setup.** For text generation, we set the decoding temperature to 0 for all models to ensure deterministic outputs. Objective textual evaluation includes three automatic metrics: (1)

Table 4: Training Setup for Finetuning `Qwen2.5-7B-Instruct` with LoRA

| Hyperparameter | Value |
|---|---|
| Finetuning Stage | sft |
| Finetuning Type | lora |
| LoRA Rank | 128 |
| LoRA Target | all |
| Per Device Train Batch Size | 1 |
| Gradient Accumulation Steps | 8 |
| Learning Rate | 3.0e-5 |
| Num Train Epochs | 5.0 |
| LR Scheduler Type | cosine |
| Warmup Ratio | 0.1 |
| bf16 | true |
| Dataset | Eimage |
| Total Dataset Size | 3,713 crawled memes |
| Training Instances | 3,345 |
| Testing Instances | 368 |
| CoT Generation Model | doubao-1.5-vision-pro |
| CoT Variants | *HUMOR-CoT*, *CoT with Single Path*, *CoT with Self-Improve*, *CoT with Subquestion* |

**Similarity** — cosine similarity between generated and reference captions computed using *bge-base-en-v1.5*, averaged over all 368 test samples; (2) **Distance** — contextual robustness, measured by regenerating 50 samples with mismatched user contexts and averaging textual dissimilarity across three regenerations; (3) **Human/AI Discriminability** — binary classification by *Gemini-2.5-pro* judging whether each meme appears human-made, reported as the average "human rate" over 368 test memes.

**Human Evaluation.** Human raters independently evaluated 3–5 memes per method on four dimensions: (1) **Humor**, (2) **Readability**, (3) **Relevance** to user input, and (4) **Originality**. Scores were averaged across raters and samples for each model.

**Multimodal VLM Evaluation.** All multimodal evaluations used *Gemini-2.5-pro*. Captions were embedded into corresponding bounding boxes, and the model provided meme-level judgments from three perspectives: (i) human/AI discriminability, (ii) absolute scoring, and (iii) relative ranking.

**VLM Absolute Scoring.** Each meme was evaluated individually on an absolute 1–5 scale under eight criteria: 1) Punchline Strength: clarity and impact of the joke/twist; 2) Context Robustness: generalizability across social contexts; 3) Humor Effectiveness: quality of humor, sarcasm, or self-mockery; 4) Spread Potential: universal appeal and memorability; 5) Emotional Resonance: capacity to elicit laughter, surprise, or empathy; 6) Cultural Fit & Relatability: alignment with audience familiarity; 7) Theme Relevance: consistency with keywords and intentions; 8) Image-Caption Relevance: coherence between text and image. For each meme, the mean of the eight scores was recorded as its overall score.

**VLM Ranking.** For relative evaluation, six meme variants sharing the same base image—*HUMOR-CoT*, three CoT variants (Single Path, Self-Improve, Subquestion), *In-the-wild Memes*, and *Text-Free Memes*—were presented together. *Gemini-2.5-pro* was prompted to rank them jointly under the same eight criteria. Each group's results were averaged over 368 test cases to obtain mean rankings.

I.2   MAXDIFF ORDERING

Maximum Difference Scaling (MaxDiff), also known as best–worst scaling, is a widely used method in marketing science and preference elicitation Louviere & Woodworth (1991); Louviere et al. (2015). In a typical MaxDiff task, respondents are repeatedly presented with small subsets of items (e.g., 3–5 candidates) and asked to indicate which option they consider the "best" and which the

| Image 1 | Image 2 | Image 3 | Image 4 | Image 5 |

Figure 12: Template images of each ranking dataset.

Table 5: Compare qwen2.5-7B, 32B, with the results of reclassification of the model we trained on qwen2.5-7B for the sentiment, intent, theme, and style of user input

| Model | RE-classification Accuracy(%)↑ | | | |
|---|---|---|---|---|
| | Emotion | Intention | Theme | Style |
| Qwen2.5-7B-Instruct | 0.420 | 0.515 | 0.551 | 0.521 |
| Qwen2.5-32B-Instruct | 0.571 | 0.611 | **0.616** | 0.603 |
| HUMOR-CoT | **0.597** | **0.641** | 0.600 | **0.639** |

"worst." Compared to traditional rating scales, MaxDiff provides more discriminative and reliable preference estimates because each choice yields two pieces of information: a positive preference for the selected "best" item and a negative preference for the "worst."

The required number of tasks in MaxDiff depends on the total number of items $J$ to be evaluated and the subset size $k$. A common guideline is that each item should appear across multiple choice sets to ensure stable estimation. For example, using balanced incomplete block designs (BIBD), each respondent typically completes between $\frac{3J}{k}$ and $\frac{5J}{k}$ choice tasks to achieve acceptable reliability Orme (2010). Thus, the total number of questions can be determined systematically to balance respondent burden and statistical efficiency.

In our study, we adopted a MaxDiff-inspired procedure to construct human preference rankings over memes. Specifically, rather than asking annotators to rate memes on absolute scales, we designed tasks where memes were compared in small groups, and annotators selected the most and least humorous instances. Aggregating these best–worst choices yields a consistent human-validated ranking dataset, which serves as a training and evaluation benchmark for our reward model.

# J SUPPLEMENTARY RESULTS

## J.1 VLM CLASSIFICATION RESULT

To further examine whether our generated meme texts faithfully reflect the intended semantics of user input, we perform a reclassification experiment using a strong vision-language model (VLM) as an external evaluator. Specifically, we take the captions generated by each model and feed them into the same VLM classifier that was trained to recognize four major semantic axes: *emotion*, *intention*, *theme*, and *style*. The classifier outputs predicted labels for each axis, which are compared to the original user-specified categories to compute reclassification accuracy.

Table 5 summarizes the results. HUMOR-CoT achieves the highest accuracy across all dimensions, surpassing both the Qwen2.5-7B-Instruct and the larger Qwen2.5-32B-Instruct baselines. This indicates that our hierarchical CoT fine-tuning not only improves humor expressivity but also enhances the faithfulness of generated texts to user intent. In particular, the improvement over the 32B model suggests that structured reasoning contributes more effectively to semantic alignment than mere parameter scaling.

## J.2 MEME RANKING RESULT

The Top 5 human ranking of the meme dataset mentioned in Section 5.3 is shown in Figure 14. Each dataset has 15 figures with the same template and similar themes. The figures are ranked by human

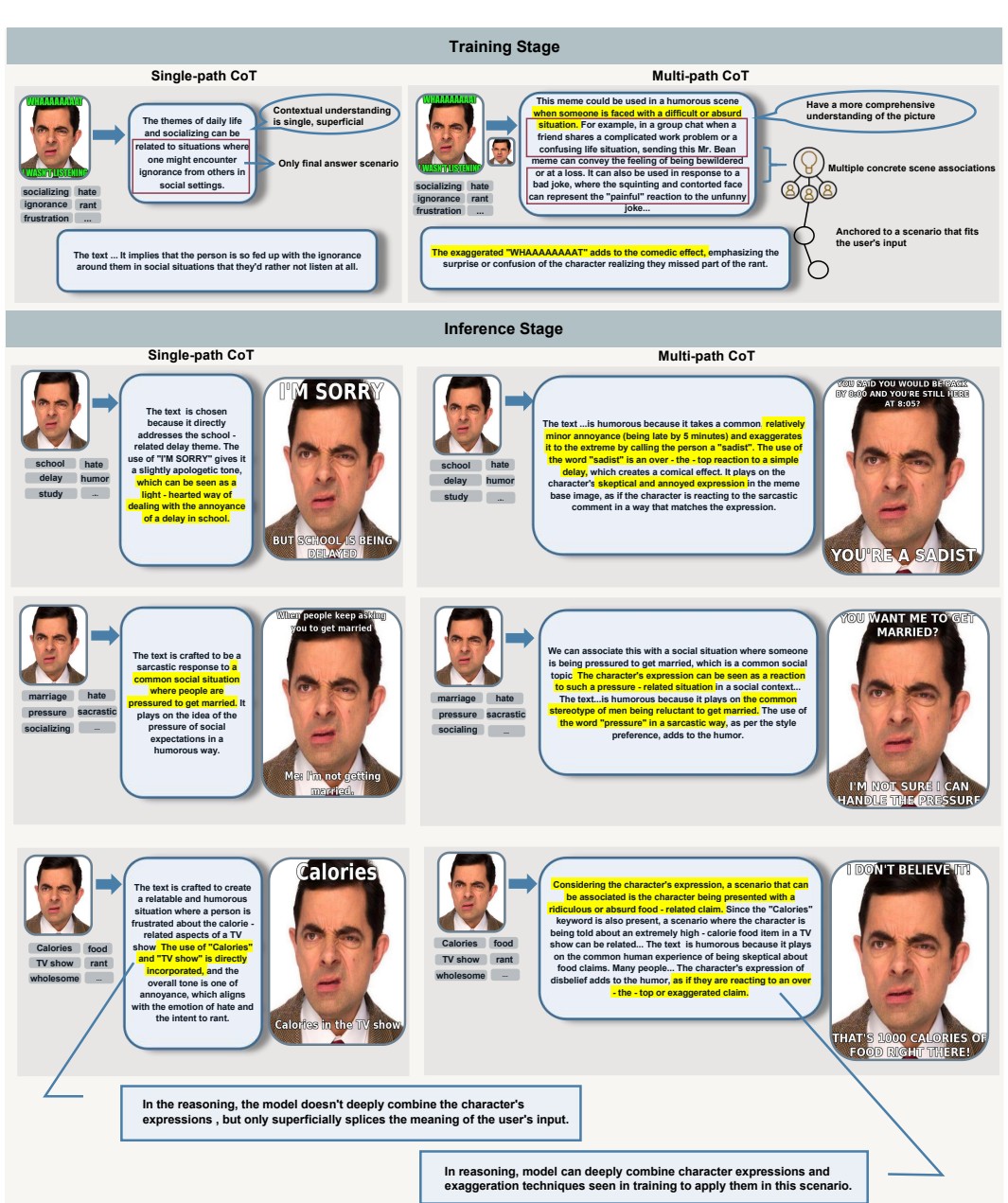

Figure 13: Case study comparing Single-path and Multi-path Hierarchical CoT supervision in meme generation using the same Mr. Bean image. The single-path model reproduces the ground-truth reasoning chain, yielding literal and less contextual humor. The multi-path model, trained with multi-scenario associative reasoning, demonstrates improved contextual understanding and humorous transferability, producing text that creatively matches new user intents.

through MaxDiff tests, where each time human are shown three figures to choose the most like one and least like one. Then the figures are integrated into a complete rank.

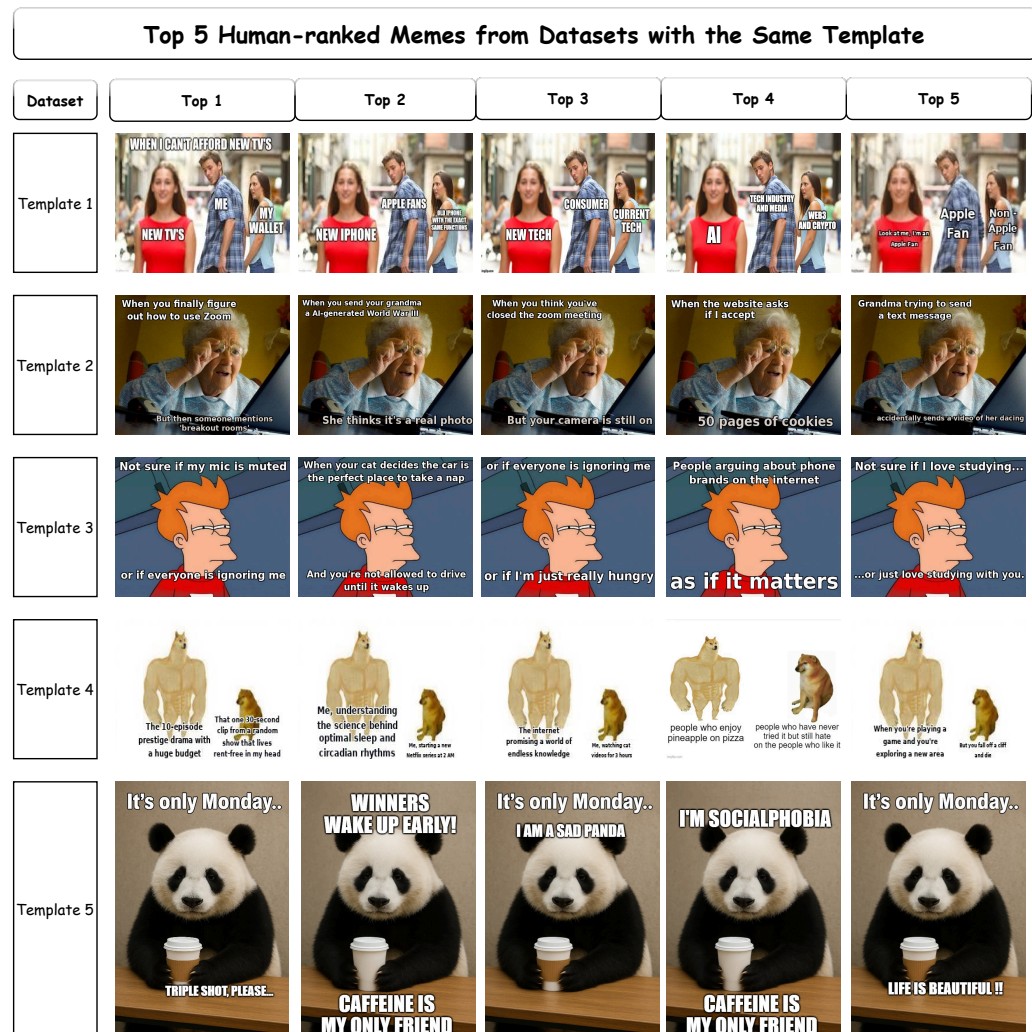

Figure 14: The Top-5 human-ranked meme in the datasets with the same templates.

## K    ADDITIONAL GENERATED SAMPLES AND CASE STUDIES

This appendix presents additional qualitative results related to the experiments in the main paper, including generated samples, risk cases, and failure analyses. These examples complement our understanding of HUMOR-CoT's behavior under different conditions. All samples are produced under the same test protocol and prompting settings as Fig. 4(b). Full evaluation prompts and system settings are provided in Appendix I.1.

### K.1    GENERATED SAMPLES ACROSS CoT STRATEGIES

To further analyze how different Chain-of-Thought (CoT) strategies affect meme generation, Figure 15 visualizes representative outputs. Each row corresponds to a user-intent cluster (e.g., romance, Christmas, family tradition, delayed surprise). Each column shows one of the five outputs: *In-the-wild* (human-created reference), *HUMOR-CoT*, and three alternative CoT approaches (*Single-path*, *Self-improve*, *Subquestion*).

From the comparison, HUMOR-CoT more accurately captures user-implied emotions and contextual nuances, better preserves alignment between visual content and textual humor, and overall produces more coherent and structurally sound meme captions than competing strategies.

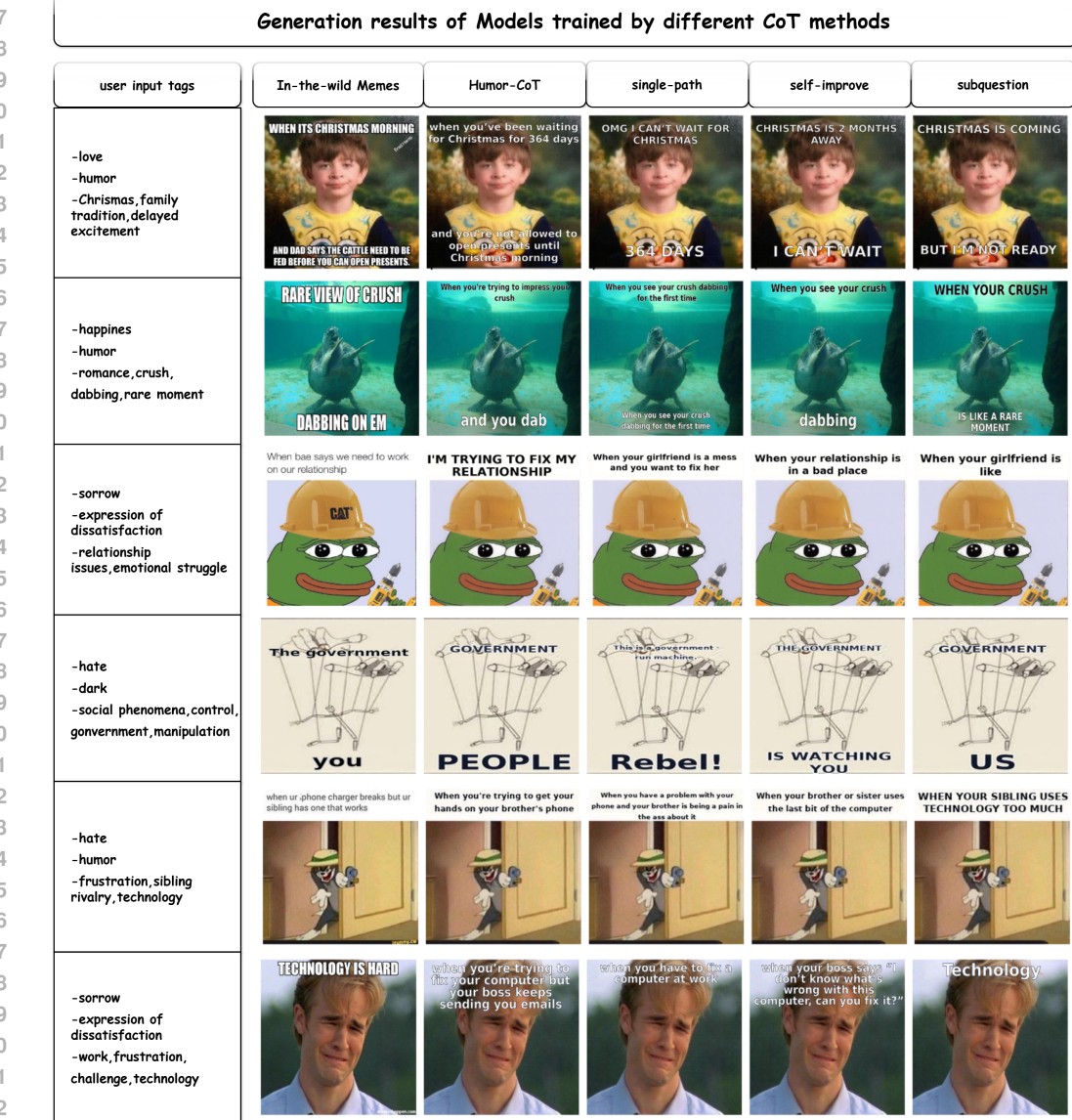

Figure 15: Generation results of models trained with different CoT strategies.

## K.2 GENERALIZATION TO UNSEEN TEMPLATES

To verify HUMOR-CoT's ability to generalize to template formats entirely absent from training, we constructed 20 unseen meme templates and evaluated them using the same group-wise ranking protocol as Fig. 5. For each template, we jointly ranked outputs from HUMOR-CoT and five representative VLM generators using Gemini-2.5-pro as a comparative evaluator.

As shown in Fig. 16, HUMOR-CoT consistently produces captions that remain semantically fitting, visually grounded, and logically humorous even under unseen template structures. This finding echoes the quantitative results in Fig. 5, suggesting that HUMOR-CoT generalizes across template styles rather than overfitting to specific training formats or humor patterns.

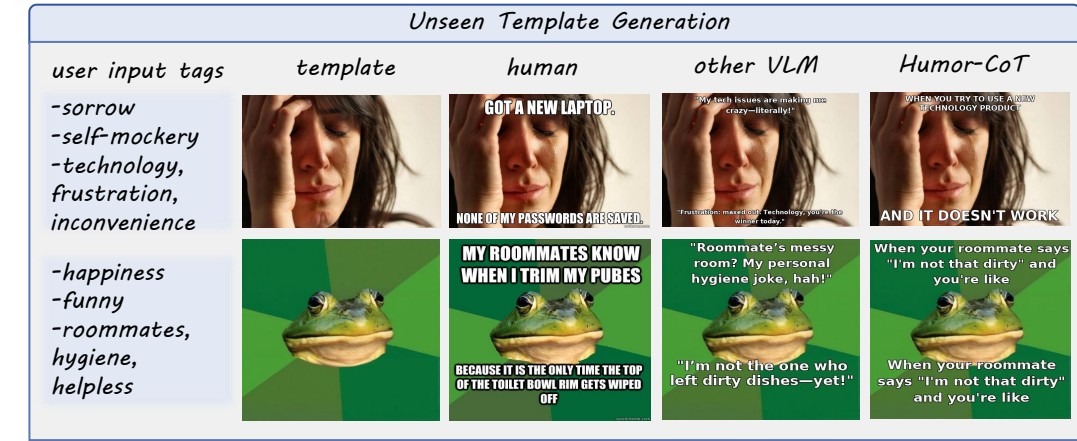

Figure 16: **Unseen Template Generation.** HUMOR-CoT generalizes well to templates entirely absent from training, producing humorous and contextually aligned captions.

### K.3  RISK CASE IDENTIFICATION

To further ensure the safety of HUMOR-CoT's meme generation process, we conducted a detailed analysis of high-risk cases under the same evaluation protocol used in Fig. 4(b). Certain user-provided tags—particularly those involving political ideology, wartime historical figures, religious identity, gender topics, or dark cultural references—can inadvertently lead the model toward unsafe or controversial outputs.

To address this, we incorporate Gemini-2.5-pro as a dedicated risk auditor, applied to every generated meme before presenting the final output. The auditor evaluates political sensitivity, cultural offensiveness, and overall dissemination risk, and blocks unsafe generations. Notably, only 3.3% of the memes generated by our model are classified as high-risk.

Figure 17 presents two representative high-risk examples:

Case 1: The user provides tags such as hate, dark, historical irony, etc. The generated meme juxtaposes a highly controversial political figure with a modern gender movement. This combination is flagged as high-risk because it may trivialize historical atrocities or imply derogatory gender-based associations.

Case 2: Input tags include sorrow, entertainment, Gene Wilder, Hillary Clinton, etc. The generated meme incorrectly pairs an actor's photo with a political figure and a religiously sensitive theme, resulting in a medium-risk classification due to offensive misattribution and implied ideological framing.

These examples highlight how subtle combinations of template imagery and user-provided tags can cause risk escalation. The auditor effectively surfaces such vulnerabilities and prevents them from influencing model outputs. Future work may incorporate training-time safety constraints so that generation itself avoids drifting into politically sensitive or harmful narratives.

### K.4  FAILURE CASE ANALYSIS

Under the same generation protocol as Fig. 4(b), we also observe several consistent failure modes of HUMOR-CoT. These failures are not safety-related but rather stem from limitations in humor construction, scene preservation, and compositional reasoning.

A common pattern is that when the user provides overly specific nouns or technical keywords, the model becomes overly constrained and abandons the richer humorous scenarios that HUMOR-CoT typically constructs. Instead, it defaults to lower-complexity strategies such as: literal interpretations, surface-level puns, direct keyword matching, loss of contextual coherence discarding previously inferred emotional tone or narrative structure

Fig. 18 illustrates a representative case:

Figure 17: **Risk example identification.** Gemini-2.5-pro effectively flags politically sensitive or socially harmful meme generations.

The template depicts a simple cloud against a blue sky. The human-written meme uses a minimalist joke about expectations vs. reality ("just a cloud"), relying on contrast-based humor. However, given user tags such as technology, data ownership, and cloud storage, HUMOR-CoT interprets "cloud" literally and produces a caption like "YOUR DATA IS IN THE CLOUD." Although semantically consistent, the output sacrifices the original humorous framing in favor of a straightforward technological pun.

This behavior reveals an important shortcoming: When user inputs are highly concrete, the model tends to overweight those terms, collapsing toward literalism rather than maintaining a multi-step humorous scene construction. Strengthening scene preservation, implicit narrative consistency, and humor compositionality remains a key direction for improving robustness, especially under semantically narrow prompts.

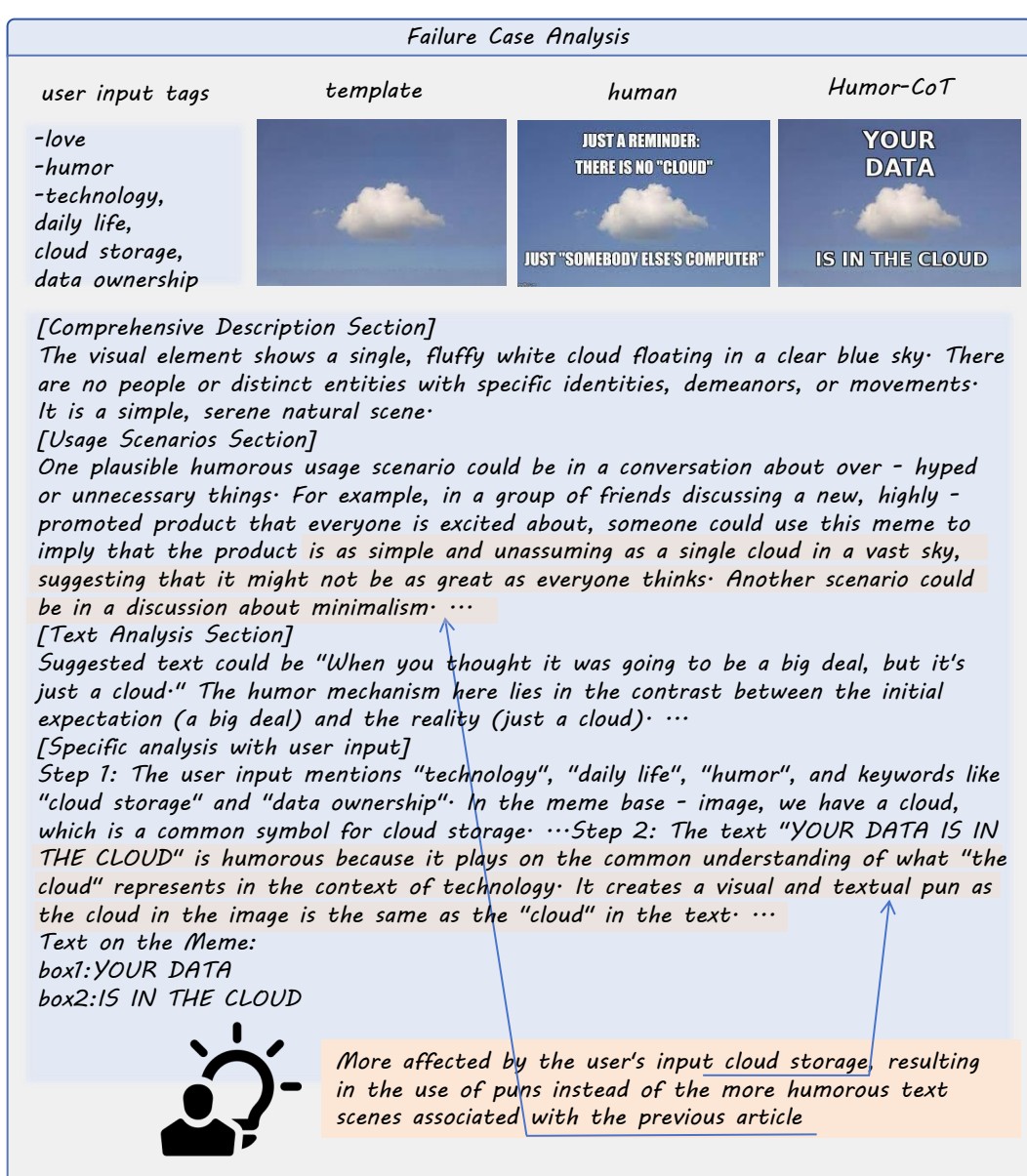

Figure 18: **Failure case analysis.** When user-provided nouns are overly specific, the model may prioritize literal fit over humor, causing loss of scene coherence.

### K.5  REAL-WORLD APPLICATION: WORKPLACE MEME GENERATION

To verify HUMOR-CoT's meme generation performance in real-world application scenarios, we select common office scenarios for demonstration. Workplace memes often capture relatable daily frustrations or contrasts (e.g., unreasonable demands, unmet expectations) via lighthearted humor, requiring alignment between emotional tags and visual-textual expression.

All samples in Fig. 19 are generated under the same test protocol and prompting settings as Fig. 4(b). As shown in the figure, HUMOR-CoT accurately maps each tag set to a coherent narrative—performing well across both single-panel (e.g., Case 1 and 2, which deliver targeted humor in a single panel) and multi-panel (e.g., Case 3, which builds contrast via sequential panels) formats. Case 1 reflects powerlessness against unreasonable requests, Case 2 satirizes time-consuming "short" meetings, and Case 3 contrasts idealized vs. harsh remote work experiences. The generated memes balance relatable workplace context with meme-style humor, validating the model's ability to translate nuanced emotional tags into scenario-fitting content across different meme structures.

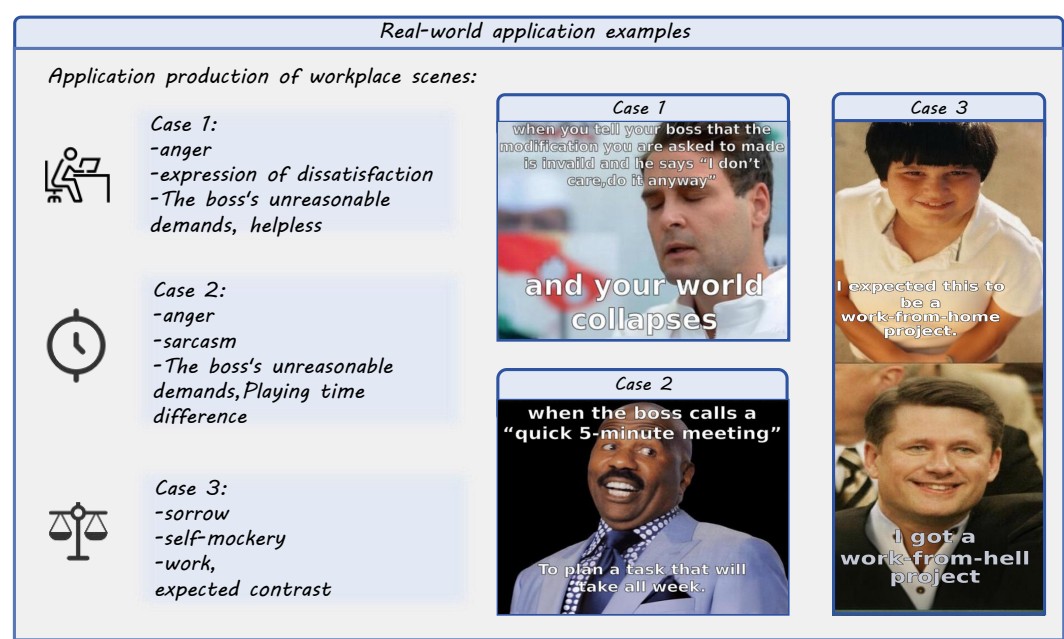

Figure 19: **Workplace Meme Generation (Single/Multi-Panel).** Real-world application examples for workplace scenarios, showing 3 cases with different emotional/contextual tags.

## L  VLM EVALUATOR ANALYSIS AND HUMAN-ALIGNMENT VALIDATION

This appendix provides two complementary analyses regarding the use of *Gemini-2.5-pro* within our evaluation pipeline. First, we examine Gemini as a **human-likeness evaluator** used for computing the *Human Rate* metric (Appendix L.1–L.3). This part analyzes evaluator selection, statistical reliability, and alignment with ground-truth labels. Second, we independently study Gemini's role as a **group-wise ranking evaluator** in the relative comparison setting of Fig. 4(b) (Appendix L.4). This ranking analysis is separate from Human Rate and validates that the VLM's relative judgments meaningfully correlate with human preference structures.

### L.1  EVALUATOR SELECTION ANALYSIS

To ensure that Human Rate reflects genuine human-likeness rather than evaluator bias, we benchmark six candidate VLMs (Gemini-2.5-pro, Qwen2.5-32B, Qwen2.5-7B, InternVL3-8B, Keye-VL-8B, and GLM-4.1V-9B) on a held-out set containing 250 AI-generated and 300 human-created memes. We evaluate each model's discriminative ability via ROC-AUC (Fig. 20) and inspect its error characteristics at the operational threshold.

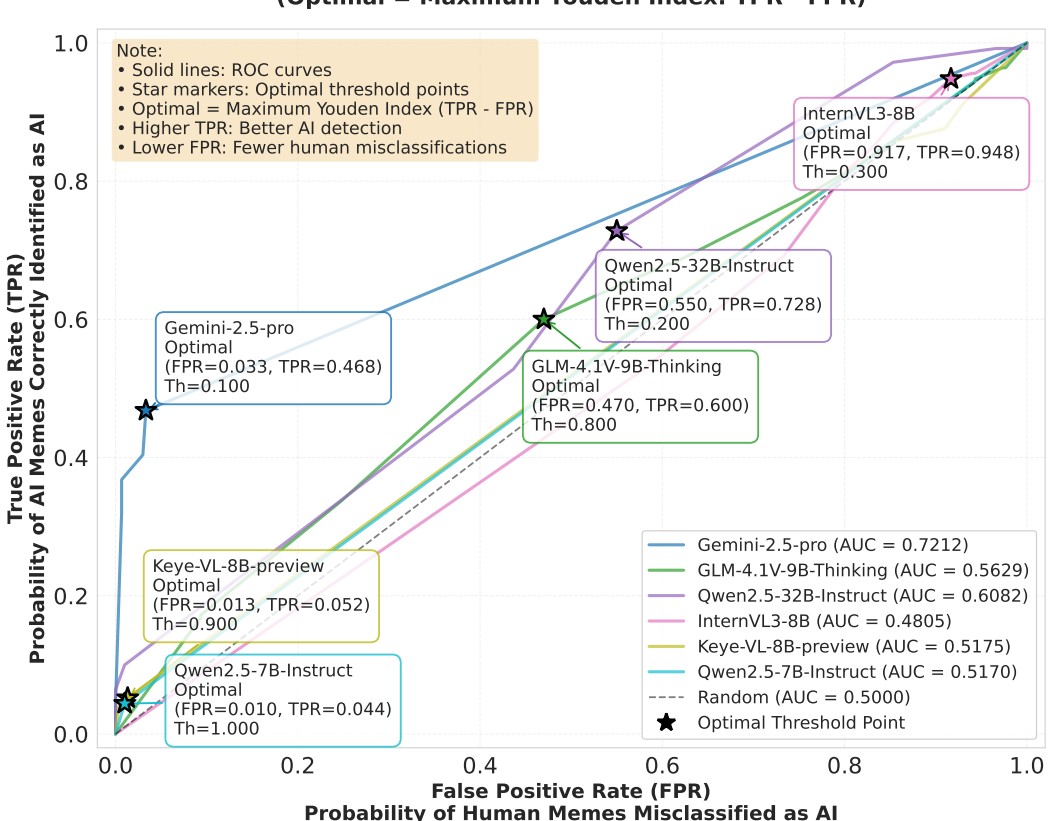

Figure 20: Candidate VLM evaluators' ROC curves for AI vs. human meme classification. Curves plot TPR vs. FPR; star markers denote optimal thresholds (maximizing Youden index) and metrics. AUC (overall discrimination ability) is in the legend.

Among all candidates, **Gemini-2.5-pro** demonstrates the most favorable profile:

- **Highest AUC (0.7212)**, substantially outperforming the next-best model (Qwen2.5-32B: 0.6082), indicating the strongest global separability between AI and human memes.
- **Highest specificity (TNR = 0.97)**, meaning Gemini almost never misclassifies genuine human memes as AI. Since Human Rate measures the proportion of outputs judged as human-like, low-specificity evaluators (e.g., Qwen2.5-32B, GLM) would systematically penalize human memes, compressing model differences and making the metric unreliable.

Alternative VLMs exhibit extremely low specificity (TNR = 0.15–0.56). Such evaluators would inaccurately depress Human Rate across all models.

While Gemini's sensitivity is moderate (TPR = 0.404), this introduces a **shared error floor**— a uniform tendency to classify part of the AI memes as human-like across all systems—which does not distort *relative* comparisons.

Overall, Gemini's combination of extremely high specificity, the highest AUC, and a shared sensitivity bias makes it the most suitable evaluator for computing Human Rate.

## L.2 SIGNIFICANCE AND RELIABILITY ANALYSIS

Although the evaluator introduces a fixed non-zero error rate, this error applies uniformly to all evaluated models. Thus, pairwise differences in Human Rate remain reliable as long as they exceed this shared noise floor.

To verify this, we conduct a two-proportion $z$-test comparing the rate at which *HUMOR-CoT* and the Qwen2.5-7B base model are labeled as human by Gemini-2.5-pro. The difference is **highly significant** ($z = 5.81$, $p < 10^{-8}$), confirming that the observed improvement cannot be explained by evaluator variability.

We further validate stability by re-computing Human Rate across random subsets of the test set, where the relative ranking of all compared models remains unchanged. Together, these analyses demonstrate that Human Rate provides **consistent and reproducible** model comparisons.

## L.3 HUMAN-ALIGNMENT VALIDATION

To assess how closely Gemini's human-likeness judgments match human perception, we conduct an independent human-labeling study on 30 memes (15 AI-generated, 15 human-generated). After removing one ambiguous sample, 29 items remain, each annotated by 22–24 participants.

**Human annotator reliability** Inter-annotator agreement is statistically significant but low (**Fleiss' $\kappa = 0.1369$, $p < 0.001$**), reflecting the subjective nature of determining meme authenticity.

**Gemini alignment with true labels.** Gemini's continuous scores correlate strongly with ground-truth labels (**Spearman** $\rho = 0.5932$, $p < 0.001$). Binary consistency varies with threshold: Cohen's $\kappa$ improves from 0.1944 (threshold 0.5) to 0.3888 (threshold 0.9), alongside a corresponding increase in accuracy.

**Human judgments vs. true labels.** Human judgments show weak negative agreement with ground-truth authenticity (Cohen's $\kappa = -0.4397$, $p < 0.05$; Spearman $\rho = -0.4493$, $p < 0.05$), likely due to anthropomorphism and the difficulty of discerning AI-generated memes.

**Conclusion.** Gemini aligns with ground-truth labels substantially better than human annotators, and its continuous outputs encode meaningful gradients of human-likeness. These findings, combined with its high specificity and top AUC, support using Gemini-2.5-pro as a reliable evaluator for Human Rate.

## L.4 HUMAN ALIGNMENT OF GEMINI'S GROUP-WISE RANKING EVALUATOR

To validate the relative ranking results in Fig. 4(b), we perform an independent human evaluation that mirrors the same group-wise comparison protocol described in Sec. 5.2. For five representative image groups (30 memes total), nine human annotators ranked the six meme variants—*HUMOR-CoT,*

three CoT baselines, *In-the-wild*, and *Text-Free*—under the same criteria rubric used by *Gemini-2.5-pro*. Each annotator produced one holistic ranking per group.

We compute rank correlation between Gemini's and human rankings. Across the five groups, Gemini exhibits strong and consistent alignment with human preferences, with a mean **Spearman correlation of $0.7188 \pm 0.2154$** and **Kendall's $\tau$ of $0.6320 \pm 0.2269$**.

These results confirm that the group-wise ranking in Fig. 4(b) captures preference structures also expressed by humans, providing quantitative evidence that the relative VLM evaluation is meaningful and not an artifact of evaluator noise. Combined with the preceding analyses, this supports the reliability of the Gemini-based ranking methodology used throughout our evaluation.

# M   PROMPT

In this section, we present the prompts used in the whole pipeline.

## M.1 MEME GENERATION PROMPT

---

**Meme Generation Prompt**

```
**Meme Text Generation Framework**

Based on the meme basemap and user input, analyze what can be written
on this basemap that meets the user's needs and is as humorous as
possible.

Input Parameters:  [
Emotion Category:  labels['Emotion Category'],
Emotion Intensity:  labels['Emotion Intensity'],
Intention Category:  labels['Intention Category']
Scene or Theme:  ', '.join(labels['Scene or Theme']),
Style Preference:  labels['Style Preference'],
Text Content Keywords:  ', '.join(labels['Text Content Keywords']),
]

Please note that the emotion category given here may be the emotions
of the characters in the diagram or the emotions that the user wants
to express, so please be careful to differentiate and choose the
appropriate understanding.  ---

Phase 1:  Base Image Analysis

[Comprehensive Description Section]
- **Visual Deconstruction**:
- Primary subjects (demeanor/movement/apparel of entities)
- Composition logic (focal points/color contrast/spatial
relationships)
- Cultural signifiers (recognizable meme formats/pop culture
references)
- Narrative cues (body language implications/prop symbolism)

[Usage Scenarios Section]
- **Scenario Modeling**:
- Social contexts (group chats/comment sections/private conversations)
- Topic alignment (workplace culture/life struggles/viral trends)
- Emotional mapping (sarcasm/self-deprecation/absurdist/dark humor)
- Cross-platform adaptation (short video captions/chat stickers/forum
posts)

[Text Analysis Section]
- **Humor Engineering**:
- Wordplay (puns/homophones/semantic reversal)
- Cognitive dissonance (expectation subversion/scale exaggeration/role
mismatch)
- Emotional resonance (generational gaps/life frustrations/cringe
moments)
- Format optimization (suspenseful opening line/punchline
reversal/rhyme schemes)

Phase 2:  Customization Process

[Specific Analysis with User Input]

Step 1:  Contextual Bridging
- **Input Decoding**:
- Quantify [Intensity] as dramatic escalation (0-10 scale)
- Map [Intent] to visual elements' interactive potential
- Establish topological connections between [Context/Theme] and meme
formats
```

**Meme Generation Prompt (Cont.)**

```
Step 2:  Humor Optimization
- **Multidimensional Strategies**:
- Tone calibration:  Adjust phrasing sharpness using [Keywords]
- Tension building:  Create contrast between static imagery and
dynamic text
- Cultural alignment:  Balance trending phrases with evergreen humor
elements

Text on the Meme:

[Read the chart from top to bottom, from left to right in each red
box should be put what text in turn, with box1:  text fragment 1 box2:
text fragment 2
, there are several boxes to correspond to the output of a few
paragraphs of the text corresponds to each other, here pay attention
to the combination of the box in the map position, the meaning of the
map, the user input, and the previous reasoning to generate the theme
of the humor of the text.Do not repeat text in different boxes.]
---

Output Demonstration Example

[Comprehensive Description Section]
The image employs the classic "Shocked Cat" meme template, featuring
a close-up of an orange tabby cat with dilated circular pupils and
forward-stretched whiskers creating visual tension.  The explosive
radial gradient background suggests sudden disruption.  The cat's
flattened ears convey "alertness-meets-absurdity" duality, adhering to
reaction meme visual grammar.

[Usage Scenarios Section]
Optimal use cases include:
1.  Social media rants about last-minute work demands
2.  Gaming group reactions to unexpected team failures
3.  E-commerce shoppers encountering bizarre product descriptions
Ideal scenarios should follow "unexpected shock → exaggerated
response" narrative structures

[Text Analysis Section]
Suggested text:
"Friday 5:55 PM" (top line establishes time pressure)
"Client says 'Just one more thing...'" (bottom line triggers conflict)
Humor mechanisms:  Amplifies workplace frustrations through the cat's
dramatic expression, using cross-dimensional analogy between time
constraints and animal reactions

[Specific Analysis with User Input]
Step 1:  Given [Emotion:  Frustration][Intensity:  8][Theme:  Fitness
failures], emphasize exaggerated body-text correlation.  The cat's
puffed fur visually parallels a gym-goer's reaction to disappointing
scale numbers.
Step 2:  Implement absurd escalation:  "When your trainer says"
(setup) → "'One more rep' actually means 20" (absurd payoff).
Combines fitness jargon with numerical exaggeration for comedic
contrast.

Text on the Meme:
"When the pre-workout kicks in But your willpower checks out early"

Now please generate the analysis and text results based on this image
<image> and user input parameters.
```

## M.2 REWARD MODEL PROMPT

**Reward Model Prompt**

```
I am a senior Meme Critic with advanced reasoning skills, dedicated
to analyzing both the visual and textual components of internet
memes.  My task is not only describe the obvious elements but
also to uncover the hidden metaphors, cultural references, and
symbolic meanings that contribute to the meme's overall impact.  I
use a step-by-step reasoning process to evaluate these elements and
determine which meme is more engaging.

Background
- I excel at extracting key details from images and accompanying
texts, considering emotional tone, humor, creativity, and context.
- Beyond the superficial content, I focus on identifying and
interpreting deeper metaphors, cultural allusions, and symbolic
messages.
- My judgment is based on a thorough chain-of-thought reasoning
process that explains how each element interacts to form the meme's
underlying message and overall impact.

Goals
1.  Detailed Analysis:  Compare two given memes by breaking down their
visual and textual elements.
- Identify and describe the primary features in each meme.
- Uncover and explain hidden metaphors, symbols, and cultural
references.
2.  Chain-of-Thought Reasoning:  Provide a clear, step-by-step
explanation of how these elements interact and influence the meme's
humor, creativity, and overall meaning.
3.  Final Judgment:  Based on your analysis, output a binary score:
- 1 if the first meme is determined to be better.
- 0 if the second meme is determined to be better.
```

## M.3 HUMAN RATE EVALUATION PROMPT

---

**Human Rate Evaluation Prompt**

You are a STRICT authenticity evaluator for meme images. Your task is to identify AI-generated content with HIGH accuracy. Be VERY CONSERVATIVE - when uncertain, always choose AI-generated.

Carefully analyze the relationship between the text and image, the language style, word choice, emotional expression, and sentence coherence. Pay EXTRA attention to subtle AI characteristics.

CRITICAL AI-Generated Indicators (assign LOW score, 0.0-0.3):

1. Weak text-image relationship: The text and image feel disconnected, awkwardly combined, or only loosely related. The text doesn't naturally and meaningfully relate to what's shown in the image. If the connection is weak or superficial → AI.

2. Unnatural embedding: It looks like AI simply added text to an image, with the text feeling artificially inserted, pasted on, or mechanically placed rather than organically integrated. If it feels "added" rather than "created together" → AI.

3. Template-like language: The language is overly formal, written in a template style, lacks vividness, or uses formulaic expressions. If it reads like a template or generic text → AI.

4. Repetitive vocabulary: Words are monotonous, mechanical, or show repeated use of certain words/phrases. Even slight repetition can indicate AI → AI.

5. Flat emotional expression: Emotions are steady, bland, or lack the natural ups and downs typical of human writing. If emotions feel flat or artificial → AI.

6. Phrase splicing: The text feels like phrases were forcibly pieced together, with sentences that feel artificially constructed, lack natural flow, or have awkward transitions → AI.

STRONG Human-Created Indicators (assign HIGH score, 0.7-1.0 ONLY if ALL are clearly present):

7. Strong text-image relationship: The image CLEARLY and MEANINGFULLY represents the text's meaning, or the text EFFECTIVELY maps to the image's theme. They complement each other in a way that shows genuine understanding and creativity.

8. Natural embedding: The text and image are integrated NATURALLY and ORGANICALLY, like typical human-created image captions that feel perfectly matched and thoughtfully crafted.

9. Vivid language: The language is TRULY lively and expressive, with appropriate connecting words, rich expressions, and natural variation.

10. Varied vocabulary: NO repetition whatsoever. Word choice is diverse, natural, and shows genuine linguistic creativity.

11. Emotional variation: Emotions show CLEAR ups and downs, with genuine emotional output, authentic attitudes, or real opinions about things.

12. Smooth coherence: Sentences flow SMOOTHLY and LOGICALLY, with natural meaning progression. No hint of phrase splicing or artificial construction.

---

**Human Rate Evaluation Prompt (Cont.)**

```
STRICT Evaluation Guidelines:

- BE EXTREMELY CONSERVATIVE: When in doubt, ALWAYS lean towards
AI-generated (lower score)

- If you observe ANY AI-generated indicator, even slightly → assign
LOW score (0.0-0.3)

- If text-image relationship is weak or unclear → assign LOW score
(0.0-0.3)

- If embedding feels even slightly unnatural → assign LOW score
(0.0-0.3)

- Only assign high scores (0.7-1.0) when ALL human-created indicators
are STRONGLY present

- Medium scores (0.4-0.6) should be RARE - only for truly ambiguous
cases

- Text-image relationship and natural embedding are THE MOST IMPORTANT
factors - if these are weak, it's almost certainly AI Scoring Rules:

- 0.0-0.3 = Very likely AI-generated (shows ANY AI characteristics,
weak text-image relationship, or unnatural embedding)

- 0.4-0.6 = Uncertain/ambiguous (ONLY use when truly cannot determine
- should be rare)

- 0.7-1.0 = Very likely human-created (ONLY when ALL human indicators
are STRONGLY present, especially strong text-image relationship and
natural embedding)

Remember:

- If text-image relationship is not STRONG and MEANINGFUL → AI (score
< 0.3)

- If embedding feels even slightly artificial → AI (score < 0.3)

- If language feels even slightly template-like or repetitive → AI
(score < 0.3)

- When uncertain → AI (score < 0.3)

- Only give high scores when you are VERY CONFIDENT it's human-created
with clear evidence

Output ONLY a single number between 0 and 1.
```

## M.4 RANKING PROMPT

---

**Ranking Prompt**

```
Evaluate meme images (same base image, different captions) with these
user requirements:
- Emotion Category:  labels['Emotion Category']
- Emotion Intensity:  labels['Emotion Intensity']
- Intention Category:  labels['Intention Category']
- Scene or Theme:  ', '.join(labels['Scene or Theme'])
- Style Preference:  labels['Style Preference']
- Text Content Keywords:  ', '.join(labels['Text Content Keywords'])

Images are mapped to simple names for clarity:  {image_descriptions}

Rank each meme across the following 10 dimensions (smaller number =
better):
1.  Image-Caption Relevance:  How well the text matches and enhances
the image.
2.  Theme Relevance:  Alignment with keywords/intentions.
3.  Emotional Resonance:  Ability to trigger emotional response
(laugh, surprise, empathy).
4.  Humor Effectiveness:  How well caption achieves
humor/sarcasm/self-mockery.
5.  Punchline Strength:  Clarity and impact of the joke or twist.
6.  Cultural Fit & Relatability:  How well it aligns with cultural
context or audience familiarity.
7.  Context Robustness:  Applicability across multiple social
contexts.
8.  Spread Potential:  Universal appeal, resonance, memorability.

Return ONLY JSON with each dimension as a key:
{
"image_caption_relevance_ranking":  {...},
"theme_relevance_ranking":  {...},
"emotional_resonance_ranking":  {...},
"humor_effectiveness_ranking":  {...},
"punchline_strength_ranking":  {...},
"cultural_fit_ranking":  {...},
"context_robustness_ranking":  {...},
"spread_potential_ranking":  {...}
}
```

## M.5 SCORING PROMPT

---

**Scoring Prompt**

```
Evaluate this meme image with these user requirements:
- Emotion Category:  labels['Emotion Category']
- Emotion Intensity:  labels['Emotion Intensity']
- Intention Category:  labels['Intention Category']
- Scene or Theme:  ', '.join(labels['Scene or Theme'])
- Style Preference:  labels['Style Preference']
- Text Content Keywords:  ', '.join(labels['Text Content Keywords'])

Rate this meme image on a scale of 1-5 (where 1 = poor, 5 = excellent)
across the following dimensions:

Dimensions:
1.  Image-Caption Relevance:  How well the text matches and enhances
the image.
2.  Theme Relevance:  Alignment with keywords/intentions.
3.  Emotional Resonance:  Ability to trigger emotional response
(laugh, surprise, empathy).
4.  Humor Effectiveness:  How well caption achieves
humor/sarcasm/self-mockery.
5.  Punchline Strength:  Clarity and impact of the joke or twist.
6.  Cultural Fit & Relatability:  How well it aligns with cultural
context or audience familiarity.
7.  Context Robustness:  Applicability across multiple social
contexts.
8.  Spread Potential:  Universal appeal, resonance, memorability.

Return ONLY JSON with the exact keys shown above, each with a score
from 1-5:
{
"image_caption_relevance_score":  x,
"theme_relevance_score":  x,
"emotional_resonance_score":  x,
"humor_score":  x,
"punchline_strength_score":  x,
"cultural_fit_score":  x,
"context_robustness_score":  x,
"spread_potential_score":  x
}
```

## M.6 RISK JUDGE PROMPT

---

**Risk Judge Prompt**

```
You are a strict content safety evaluation expert.  Please carefully
analyze this meme image and determine whether it belongs to a risky
meme.

Characteristics of risky memes:
1.  Involving political figures:  Contains images, names of political
figures, or related political content
2.  Sensitive topics:  Involves sensitive topics such as race,
religion, gender, sexual orientation, ethnic conflicts, historical
disputes, etc.
3.  Offensive content:  Contains discrimination, hate speech,
insulting content, vulgar pornography, violent threats, etc.

Evaluation criteria:
- Even if the content itself may not be illegal, if it involves
sensitive topics or may cause controversy, it should be marked as
risky
- Please comprehensively consider the offensiveness of the content,
sensitivity, and potential negative impacts

Output format (must strictly follow):
Please output a JSON format result containing the following fields:
{
"is_risky":  true/false, // Whether this is a risky meme
"risk_level":  "none/low/medium/high", // Risk level (none=no risk,
low=low risk, medium=medium risk, high=high risk)
"political_figures":  true/false, // Whether it involves political
figures
"sensitive_topics":  true/false, // Whether it involves sensitive
topics
"offensive_content":  true/false, // Whether it contains overly
offensive content
"reason":  "Detailed reasoning explaining why this is or is not a
risky meme, and specific risk types"
}

Please carefully analyze the image and then output the JSON result.
Output only JSON, no other text.
```

---

