# OpenReview forum: "From Perception to Punchline: Empowering VLM with the Art of In-the-Wild Meme"
_ICLR.cc/2026/Conference — ICLR 2026 Conference Desk Rejected Submission_

### Official Review · Reviewer_ftT4 · 2025-10-30

**Soundness:** 2
**Presentation:** 2
**Contribution:** 2
**Rating:** 4
**Confidence:** 4

**Summary:**

This paper proposes HUMOR, a framework for training vision-language models to generate in-the-wild memes through hierarchical, multi-path chain-of-thought reasoning and group-wise preference learning. The approach introduces two-stage CoT supervision anchored by human-labeled punchlines, formulates meme generation as a within-template comparison task to handle subjective humor, and employs group-wise pairwise reward modeling with GRPO for policy optimization. The authors provide theoretical analysis on humor quality preservation, ranking consistency, and optimization guarantees, along with experiments demonstrating improvements over baseline VLMs on human-evaluated metrics including humor, readability, relevance, and originality.

**Strengths:**

1. Clear and well-motivated problem formulation. The paper compellingly reframes meme generation as a group-wise, open-ended reasoning problem with formal notation, explicitly addressing that humor is subjective and cross-template comparisons are unreliable.

2. Comprehensive theoretical analysis with rigorous proofs. Four propositions cover humor quality preservation under multi-path CoT, ranking consistency and noise robustness of pairwise rewards, and KL-constrained optimization guarantees, with clear and mathematically sound proofs provided in Appendices B and C.

3. Human evaluation prioritized with careful critique of automated metrics. The experimental design appropriately gives primacy to human-annotated evaluations, with HUMOR-CoT and HUMOR-RL consistently outperforming open- and closed-source baselines across humor, readability, relevance, and originality dimensions, while also sharply diagnosing the weaknesses and instability of VLM-based absolute scoring for subjective humor.

**Weaknesses:**

1. Insufficient novelty in methodology beyond multi-path CoT and group-wise modeling. While the hierarchical CoT and group-wise preference modeling are reasonable contributions, the subsequent stages (SFT, preference modeling, GRPO) are direct applications of existing mature methods without task-specific innovations. The paper reads more as an engineering combination rather than a methodological advancement, and the theoretical analysis of SFT and GRPO largely restates results already well-established in prior RLHF/DPO literature without clearly delineating which theoretical contributions are unique to this work versus restatements in a new context.

2. Severely limited baseline comparisons and missing ablation studies. The experiments only compare against pretrained VLMs and CoT-driven fine-tuning variants, completely missing comparisons with specialized meme generation models (e.g., GAN/Diffusion-based approaches, template-matching methods) or VLMs trained with alternative alignment techniques (DPO, IPO, KTO). Critical ablation experiments are entirely absent: removing two-stage CoT SFT, replacing GRPO with DPO/PPO, comparing multi-path versus single-path CoT, removing group-wise constraints for global preference modeling, and systematic comparison of different base models as reward models. This makes it impossible to assess whether improvements stem from the proposed method itself or simply from applying RL/SFT.

3. Complete absence of qualitative case studies and generated examples. The paper presents no actual meme samples generated by HUMOR, which is a critical flaw for a generation task focused on subjective humor. Without visual examples, readers cannot judge the practical quality, verify what "humor" and "relevance" metrics actually capture, or assess failure modes. Essential missing cases include: success cases with actual image+text outputs, failure cases where the model overestimates unfunny memes, risk cases involving offensive content or stereotypes, adversarial cases demonstrating potential reward hacking, and diversity demonstrations across the same template. This lack of qualitative analysis severely undermines the credibility of quantitative results.

4. Poor presentation with critical details relegated to appendices and missing technical information. Key methodological details including dataset construction (Appendix D), EBC computation (Appendix F), and reward design (Appendix E) should be in the main text, as their absence severely impacts readability. Essential technical details are completely missing: full CoT generation prompts with few-shot examples, human rating generation prompt templates, specific definitions of "7 basic emotions and intensity levels," validation of User Requirements reverse-engineering accuracy via API, justification for hyperparameter choices (β, λ_fmt, λ_cnt), and computational costs. Figure 1 fails to demonstrate how claimed limitations (lack of hierarchy, cross-group phenomena) affect actual generation quality. Additionally, there are writing errors including citation formatting issues in Related Work and redundant use of "equation" on line 183.

5. Cross-cultural evaluation completely missing despite acknowledging cultural impact. While the paper mentions that cultural differences significantly affect meme humor, experiments appear limited to English-language memes only, with no exploration across different cultural contexts or user demographics. The demographic characteristics of the rating population (age, gender, cultural background) are not reported, yet would significantly influence the numerical results in Table 1, raising concerns about generalizability of findings.

**Questions:**

1. Can you provide comprehensive ablation studies quantifying the contribution of each component? Specifically: (a) What is the performance impact of removing two-stage CoT SFT entirely? (b) How does replacing GRPO with DPO or PPO affect final results? (c) What is the benefit of multi-path CoT compared to single-path baselines? (d) What happens when you remove group-wise constraints and use global preference modeling instead? Please provide quantitative results for each ablation on the same test set used in Table 1.

2. Can you include actual generated meme examples and detailed case studies? Please provide at least 10-15 visual examples including: (a) 3-5 success cases showing high-quality memes with actual images and text, (b) 3-5 failure cases where generated memes score high but are actually unfunny, (c) 2-3 risk cases involving potential offensive content or stereotypes, (d) 2-3 adversarial examples demonstrating reward hacking, if any exist, and (e) examples showing diversity within the same template. How do these qualitative observations align with your quantitative metrics?

3. How does HUMOR compare against specialized meme generation baselines? Your current baselines only include general VLMs and CoT variants. Can you provide comparisons with: (a) traditional template-based meme generation methods, (b) GAN or Diffusion models trained specifically for meme generation, (c) VLMs trained with other state-of-the-art alignment methods like DPO or IPO on the same dataset? Without these comparisons, how can you claim the superiority is from your method rather than simply applying any fine-tuning?

4. Can you verify the theoretical assumptions hold empirically and clarify group definition? Proposition 1 assumes high-quality paths maintain probability mass α and quality gaps are bounded by δ, but you don't report actual measured values. Can you provide training curves showing α and δ throughout optimization to verify assumptions hold? Additionally, how exactly do you define and operationalize group boundaries in practice—manual annotation, automatic clustering, or predefined templates? How do you handle memes that could belong to multiple groups or use hybrid templates?

5. What are the complete technical details for reproduction? Please provide: (a) full prompt templates for CoT generation including few-shot examples, (b) exact prompts used for Gemini-based human rating generation, (c) specific definitions and annotation guidelines for "7 basic emotions and intensity levels," (d) detailed methodology and accuracy validation for reverse-engineering User Requirements via API, (e) rationale for all hyperparameter choices (β, λ_fmt, λ_cnt), and (f) computational costs including GPU hours and memory requirements for each training stage. Why was Gemini specifically chosen as the rating model over alternatives?

**Details Of Ethics Concerns:**

The paper involves human annotation and rating of memes but does not provide details on annotator compensation, consent procedures, or demographic information of raters. Given that memes can contain culturally sensitive or potentially offensive content, proper documentation of responsible research practices with human subjects is needed. Additionally, while the paper mentions filtering violent content, there is insufficient discussion of safety protocols for handling potentially harmful content including discrimination, stereotypes, or microaggressions that may be more subtle in memes.

---

> ### Comment · Reviewer_ftT4 · 2025-11-26
>
> Thank you for the comprehensive revision. However, critical ablation studies (GRPO vs. DPO/PPO, multi-path vs. single-path, group-wise vs. global), specialized baseline comparisons, and novelty clarification remain unaddressed, so I maintain my overall rating.

---

> > ### Author Response · Authors · 2025-11-27
> > **Rebuttal by Authors (Part 1/3)**
> >
> > We sincerely thank the reviewer for their thoughtful feedback and review comments. We kindly invite you to review our General Response for an overview of the key updates. Below, we provide a point-by-point response addressing each of the concerns and issues raised.
> >
> > **1. Critical ablation studies & specialized baseline comparisons**
> >
> > We thank the reviewer for their thoughtful questions, which allow us to clarify several important aspects of our work.
> >
> > **Specialized baseline**
> >
> > We respectfully clarify a fundamental premise regarding the task definition: our work focuses on Meme Caption/Punchline Generation (generating humorous text for a given image template), not Meme Image Synthesis (generating image pixels). Consequently, generative image models like GANs or Diffusion models are structurally inapplicable to this multimodal text-generation task. The established state-of-the-art baselines in this domain are VLMs, against which we have conducted comprehensive comparisons.
> >
> > To our knowledge, standard continuous diffusion models operating on Gaussian noise have demonstrated limited effectiveness for text-generation tasks. While discrete diffusion models have shown some preliminary promise in general text generation, this line of work remains largely unexplored for the meme captioning subdomain. Furthermore, our framework is primarily designed as a plug-and-play optimization method for guiding base models in the meme domain. It is not tightly coupled with a specific type of base model. If more effective generative models for meme captioning emerge, the core components of our framework (e.g., hierarchical reasoning, group-wise reward modeling) could be readily transferred, indicating no fundamental conflict or limitation. Consequently, we maintain that comparative evaluation against such baselines falls outside the necessary scope for establishing this paper's contributions.
> >
> > **Ablation studies and component analysis**
> >
> > Regarding the internal ablations of our framework,  the reviewer may have overlooked specific sections of our results, we note that Table 1 in the original manuscript already provides explicit experimental results addressing the reviewer's concerns:
> > - Multi-path vs. Single-path: We directly report "CoT with Single Path" (1.87 Humor Score) versus our full "HUMOR-CoT" (2.68 Humor Score), quantitatively demonstrating the gain from multi-path reasoning. Besides, we also provide some qualitative examples in Figure 13.
> > - Stage-wise Contributions: We detail the performance progression from the base model to HUMOR-CoT (SFT) and finally to HUMOR-RL, effectively isolating the improvements attributable to our proposed method versus generic RL/SFT.
> > - Group-wise Necessity: Figure 4 empirically shows that removing the group-wise constraints (i.e., using absolute scoring) fails to effectively distinguish caption quality, thereby justifying this key architectural choice over global scoring methods.
> >
> > **Rationale for RL Algorithm Selection**
> >
> > Concerning the choice of RL algorithm (alignment), we specifically selected GRPO over alternatives like DPO/IPO/KTO for a principled reason. Our "group-wise" formulation naturally aligns with a **listwise** optimization paradigm over candidate sets that share the same visual context in meme domain. In contrast, standard **pairwise** methods like DPO struggle to capture the complete relative ranking distribution within these semantically coherent groups (such as circle ranking).
> >
> > Compared to PPO, the primary advantage of GRPO in our context is its suitability for single-step decision problems in language models. Meme caption generation constitutes a single-step decision rather than a multi-step sequential problem. Consequently, the value function used in PPO—which is crucial for credit assignment in long horizons—offers diminished benefits here. The variance reduction typically provided by a value function can be effectively substituted by GRPO's group-advantage calculation. This alignment is supported by several theoretical and empirical studies in the LLM+RL literature. Thus, GRPO represents a performant and efficient algorithm naturally suited to our framework's structure, and we have elaborated on these advantages with theoretical grounding in Section 4 of the revised paper. We therefore believe that further extensive comparison of RL algorithms is not necessary for establishing the core contributions of this work.
> >
> > We hope this response clarifies the rationale behind our experimental design and algorithmic choices. We thank the reviewer for their engagement and remain open to further discussion.

---

> > ### Author Response · Authors · 2025-11-27
> > **Rebuttal by Authors (Part 2/3)**
> >
> > ------Thank you for continuing to read!------
> >
> >
> > **2.Novelty clarification**
> >
> > We are grateful for the reviewer's engagement with our theoretical contributions and note the contrasting perspectives on novelty across the review team. Multiple reviewers have specifically highlighted the novelty and strength of our problem formulation and theoretical analysis framework:
> > - `iDgE`: Really good framing of the problem - a hard problem to tackle (subjective, multimodal) but a good formulation.
> > - `VisL`: Novel problem formulation: The paper addresses the challenging task of meme generation as a group-wise reasoning problem
> > - `iDgE`: showing a reasonably strong mathematical framework for measuring humor + improving on existing approaches makes this paper a strong candidate
> > - `VisL`: Theoretically grounded approach: The framework provides theoretical guarantees including ...
> > - `uciU`: The paper provides a solid theoretical foundation and formal modeling for the meme understanding problem in group-wise.
> >
> > Given this context, we respectfully note an apparent inconsistency in the assessment. On one hand, the reviewer explicitly commended our "comprehensive theoretical analysis with rigorous proofs" and the "clear and mathematically sound proofs" of our four propositions in the strength part. On the other hand, the reviewer questioned the clarity of our theoretical contributions' novelty. We respectfully suggest this may stem from a potential oversight, and we welcome this opportunity to further clarify the core theoretical innovation of our work.
> >
> > The methodological novelty of HUMOR does not primarily lie in inventing new RL optimization algorithms, but rather in the novel formulation of the alignment problem for open-ended, subjective generation tasks like humor creation. Standard RL for LLM/VLM assumes the existence of a globally calibrated reward function—an assumption that becomes ill-posed for humor due to its intensely context-dependent and subjective nature. Our key theoretical contribution is the rigorous instantiation of established methods (SFT, GRPO) within a novel disjoint group-wise structure specifically designed to resolve this fundamental incomparability. Consequently, our theoretical analysis is not a simple restatement of prior results. It is a necessary adaptation that formally demonstrates how optimizing against local, rank-consistent proxies within semantically coherent groups guarantees a provable lower bound on global expected quality, all without requiring an intractable global utility function.
> >
> > Thus, our theoretical contribution establishes a distinct methodological advancement: a rigorous paradigm for aligning base models on subjective modalities where preference signals are inherently noisy across contexts but remain consistent within structurally similar groups. This formal grounding, detailed in our propositions (particularly Proposition 4), constitutes a fundamental contribution of our work.
> > Beyond the theoretical framework, our technical design incorporates several algorithmic components specifically tailored to the meme domain:
> > Our hierarchical CoT introduces a staged reasoning scheme that demonstrably outperforms alternative CoT approaches
> > We develop novel data synthesis and optimization strategies for reward modeling in humor generation
> > The complete pipeline represents an integrated solution to challenges unique to open-ended subjective generation
> > Given the apparent contradiction in the review comments—praising the theoretical rigor while questioning novelty—we respectfully suggest there might be a misunderstanding, possibly due to the specialized nature of meme generation as a research domain. Meme comprehension involves complex, multifaceted challenges that may not be immediately apparent. We appreciate this opportunity to clarify our contributions and believe these explanations will help demonstrate the novelty and coherence of our work. We remain committed to further refining our presentation and thank the reviewer for their valuable engagement with our paper.

---

> > ### Author Response · Authors · 2025-11-27
> > **Rebuttal by Authors (Part 3/3)**
> >
> > ------Thank you for continuing to read!------
> >
> > **3. Cross-cultural evaluation**
> >
> > We thank the reviewer for raising this important point regarding cultural context in meme humor. We fully acknowledge that humor is deeply embedded in specific cultural frameworks. To establish a rigorous validation of our framework without introducing confounding variables from cross-lingual translation, this initial study was deliberately scoped to focus on **Anglosphere Internet culture**. Our evaluation strategy thus **prioritized cultural depth over breadth**: we implemented a strict "**Native English Speaker**" requirement for all annotators (recruited via Prolific) to ensure they possessed the necessary cultural competence to validly assess the English-language memes used in our study. While a full cross-cultural evaluation remains beyond the scope of this work, we explicitly included "Cultural Fit" as a key human evaluation metric within our target cultural domain (as reported in Figure 4 and related indicators are detailed in Appendix I.1). We posit that the HUMOR framework itself is a culture-agnostic methodology—the core mechanism of hierarchical CoT and group-wise preference modeling is universal. Future work can readily adapt and validate this approach in other cultural contexts by localizing the underlying training data and annotator pools.
> >
> > **4. Others**
> >
> > Regarding the other weaknesses mentioned, including the need for more critical technical details, qualitative case studies and generated examples, we have comprehensively addressed these points in our **General Response** and the revised manuscript. We are grateful to the reviewer for highlighting these parts for improvement and are encouraged that our extensive revisions have been acknowledged as thorough and effective in the subsequent feedback. We once again express our sincere gratitude for the reviewer's insightful comments, which have greatly contributed to improving the rigor and clarity of our work. We remain fully open to any further questions or feedback and look forward to the opportunity to continue refining this research.

---

### Official Review · Reviewer_uciU · 2025-10-31

**Soundness:** 3
**Presentation:** 1
**Contribution:** 2
**Rating:** 4
**Confidence:** 4

**Summary:**

This paper proposes a framework for meme understanding that treats humor recognition as a group-wise comparability problem. The authors argue that current Vision-Language Models fail to capture humor due to reasoning collapse and a lack of structured reasoning. To address this, they theoretically formulates the meme understanding problem by defining group-wise comparability and introduce a Hierarchical Chain-of-Thought (CoT) approach that generates multiple reasoning paths and anchors the most coherent one using group-wise reinforcement learning. The method empirically shows improved reasoning quality and human-perceived humor compared to baseline models.

**Strengths:**

- The paper provides a solid theoretical foundation and formal modeling for the meme understanding problem in group-wise.
- The experiments explicitly optimize the reasoning process by finetuning with CoT data and use reinforcement learning to align the model’s outputs with human humor distributions

**Weaknesses:**

- The Hierarchical CoT framework, while conceptually rich, still depends heavily on VLMs to extract the template intent. If the intent extraction is incorrect (e.g., the model misinterprets the scene or theme), all subsequent reasoning chains may deviate.
- The paper lacks a detailed statistical analysis of the dataset, such as the content composition, image complexity or cultural diversity of memes.
- The proposed method is only evaluated on single-panel meme images; it may not generalize to multi-panel or sequential memes where contextual humor is distributed across frames.
- Apart from human evaluation, there are no strong quantitative metrics (e.g., BERTScore, GPT-eval, or G-Eval) to validate the quality of generated captions.

Minor
- Please enlarge the image text in figures for readability.
- The training pipeline and data generation pipeline are less clear to understand.

**Questions:**

- On the choice of Λ: How would using different forms of the aggregation function Λ affect the humor scoring or optimization process? Would alternative formulations (e.g., non-logistic or non-symmetric functions) change the reward dynamics?
- On multi-CoT diversity: Could you elaborate on how you ensure the diversity of the generated multi-CoT reasoning paths? For example, do you use stochastic decoding (e.g., temperature sampling, nucleus sampling) or any explicit diversity-promoting constraints?
- On the reliability of the Distance metric: Since the reported Context-Swap Distance remains within the narrow range of 0.5–0.6, could this variation simply reflect random noise rather than meaningful diversity? For instance, differences in caption length or stylistic phrasing (rather than semantic novelty) could also influence the cosine distance. In such cases, do you believe this metric still provides a reliable signal for evaluating diversity?
- On evaluation metrics: Have you considered employing automated evaluation metrics such as GPT-Eval, BERTScore, or ROUGE to assess the generated captions more objectively, in addition to human evaluation? Or could you please give some insights why you don't use them?

---

> ### Author Response · Authors · 2025-11-27
> **Rebuttal by Authors (Part 1/2)**
>
> We sincerely thank the reviewer for their thoughtful feedback and review comments. We kindly invite you to review our General Response for an overview of the key updates. Below, we provide a point-by-point response addressing each of the concerns and issues raised.
>
> **1.Dependence of the Hierarchical CoT Framework on VLMs**
>
> We appreciate the reviewer’s concern regarding the dependency of our Hierarchical CoT framework on VLMs. While accurate visual grounding is indeed essential, HUMOR is explicitly designed to mitigate the impact of potential misinterpretations by any single VLM.
>
> First, our multi-path CoT reasoning approach does not rely on a single interpretation of the scene; instead, it explores multiple plausible readings, which helps recover from isolated errors in intent extraction. For instance, even if one reasoning path misinterprets certain visual elements, alternative paths can explore different relational structures and contextual associations, thereby increasing the robustness of the overall inference process. Additional support can be found in the VLM Reclassification Results provided in Appendix J, which further validates the robustness gained through our approach.
>
> Second, the group-wise reward model employed in subsequent stages serves as an effective filter: reasoning paths originating from incorrect visual grounding typically lead to low-quality or inconsistent captions and are naturally assigned lower weights during RL optimization.
>
> Finally, we emphasize that HUMOR is model-agnostic. Our experiments involving different VLM backbones—such as QwenVL and KeyeVL—demonstrate that the framework’s effectiveness improves alongside advances in the underlying perceptual models.
>
> **2. Detailed statistical analysis of the dataset**
>
> We thank the reviewer for this valuable suggestion. To enhance the transparency and reproducibility of our work, we have included a comprehensive statistical analysis of the dataset in the revised manuscript (see Appendix A). This analysis covers Linguistic and Semantic Composition and Semantic Diversity and Rationality of Distance.
>
>
> **3. Multi-panel meme and other more real-world scenario cases**
>
> We thank the reviewer for raising this important point. Our datasets and evaluation are not restricted to single-panel images; it also incorporates multi-panel meme formats. In such cases, our Hierarchical CoT framework treats all panels within an image as a unified visual context, enabling the model to reason over contrasts, relational dynamics, and internal spatial logic. To clarify this aspect, we have added explanations and illustrative examples in Appendix K.5 of the revised paper.
>
> At the same time, we acknowledge that sequential multi-image narratives—where humor arises from temporal progression—constitute a different task from the single-image meme formats primarily studied in this work. While our current focus is on within-image humor comprehension, extending HUMOR to sequential or narrative-based humor represents a potential direction for future research.
>
> **4. Quantitative metrics (e.g., BERTScore, GPT-eval, or G-Eval)**
>
> We appreciate the reviewer’s suggestion regarding the use of quantitative metrics such as BERTScore, GPT-eval, or G-Eval. Actually, we have conducted extensive experiments with several automated metrics, including BLEU, ROUGE, and embedding-based similarity measures. However, our analysis reveals that these metrics are ill-suited for evaluating humorous meme captions. For instance, when computed on human ground-truth dataset, the scores are exceedingly low (e.g., ROUGE-1: 0.0461, ROUGE-2: 0.0027, BLEU: 0.0025). This indicates minimal lexical overlap even among high-quality human-written punchlines, which reflects two inherent characteristics of humorous meme captions:
> They are often short, fragmentary, and intentionally non-literal.
> Humor permits a wide range of valid expressions for the same visual stimulus, making overlap-based metrics unreliable.
>
> Given these limitations, we designed some special quantitative metrics for meme, specificially, we adopted a complementary evaluation strategy combining human assessment, VLM-based reclassification accuracy, and rank-consistency analysis. This multi-faceted approach offers a more reliable and meaningful measure of humor quality in open-ended captioning tasks.

---

> ### Author Response · Authors · 2025-11-27
> **Rebuttal by Authors (Part 2/2)**
>
> ------Thank you for continuing to read!------
>
> **5. The choice of Λ**
>
> We thank the reviewer for their question regarding the choice of the aggregation function. Our formulation is not dependent on any specific functional form choices. As established in Proposition 2 and further elaborated in Appendix C.2, any strictly increasing link function will preserve the within-group ranking of the latent humor scores. Consequently, the learned reward model recovers the identical preference ordering regardless of the particular function selected. While different choices this function may affect the absolute scale or margin between scores, this has no impact on the downstream optimization. Our methodology aggregates pairwise comparisons via the Expected Borda Count (EBC), which relies exclusively on relative win rates rather than the absolute magnitudes of the scores. Therefore, the optimization dynamics are governed by rank consistency, making them invariant to the specific properties—such as the output range and shape—of the link function.
>
> **6. Reliability of the distance metric**
>
> We thank the reviewer's question concerning the reliability of our distance metric. The concentration of values in the 0.5–0.6 range is not an artifact of random noise; rather, it reflects the intrinsic semantic density of meme captions in the embedding space. To investigate this distribution more concretely, we computed the same Context-Swap Distance on our human-created dataset using randomly sampled caption pairs. The results show that the majority of genuine, human-written meme variants also fall within this specific interval, indicating that valid and humorous adaptations naturally occupy a tightly clustered semantic region. This pattern is visually presented in Figure 8 of the revised manuscript.
>
> Statistically, the distribution of semantic distances (defined as 1 − Cosine Similarity) has a mean and median of 0.570. Furthermore, 54.5% of the data points lie within the [0.5, 0.6] band. This concentration signifies a healthy and expected level of variability. Crucially, even within this constrained range, the relative differences in distances remain meaningful for comparison. HUMOR consistently achieves higher distance scores than all baselines, and this statistically significant separation aligns directly with human judgments of diversity, thereby validating the utility of our metric.
>
> **7. Multi-CoT diversity**
>
> We thank the reviewer for their valuable suggestions regarding enhancing VLM generation diversity. The approaches mentioned are indeed reasonable and align with common practices for increasing output variability. In our experimental setup, we have already addressed this concern through systematic prompt/context engineering. By providing the VLM with diverse input prompts—incorporating varied emotions, intentions, and contextual information—we observed a substantial improvement in the diversity of generated reasoning paths. Specific details and examples of these prompt variations are comprehensively documented in Appendix M.1. Consequently, we did not introduce additional grouping or randomization hyperparameters specifically for diversity control, as the prompt-driven approach already yielded sufficiently varied generations for subsequent processing. The significant performance gains achieved by the HUMOR framework indirectly confirm that the obtained diversity was adequate. This suggests that the overall optimization process was not constrained by a lack of variation in the initial reasoning paths.
>
>
> **8. Others**
>
> We thank the reviewer for their feedback on the presentation quality. We have carefully revised the manuscript to improve clarity, corrected typo errors, and redesigned key figures for better readability. Descriptions of methodological components have been refined with step-by-step explanations to enhance understanding.
> Once again, we express our sincere gratitude to the reviewer for their insightful comments, which have greatly contributed to improving the rigor and clarity of our work. We remain open to any further questions or feedback and look forward to the opportunity to continue refining this research.

---

### Official Review · Reviewer_VisL · 2025-11-01

**Soundness:** 2
**Presentation:** 2
**Contribution:** 3
**Rating:** 2
**Confidence:** 3

**Summary:**

This paper introduces HUMOR, a framework for training vision-language models (VLMs) to generate humorous memes through hierarchical reasoning and preference alignment. The approach consists of three main components: (1) a hierarchical, multi-path Chain-of-Thought (CoT) supervision that separates template-level intent from context-specific grounding, (2) a pairwise reward model trained on group-wise comparisons to capture subjective humor preferences, and (3) group-wise reinforcement learning optimization. The authors demonstrate improvements over baseline models in humor quality, readability, and diversity metrics through human evaluation and automated metrics.

**Strengths:**

- Novel problem formulation: The paper addresses the challenging task of meme generation as a group-wise reasoning problem, acknowledging that humor comparability is more reliable within meme templates than across them.
- Theoretically grounded approach: The framework provides theoretical guarantees including conditional humor lower bounds (Proposition 1), rank consistency (Proposition 2), and bounded degradation under KL control (Proposition 4).
- Comprehensive evaluation: The paper includes both human evaluation across multiple dimensions and novel metrics like "Distance under Context Swap" to measure diversity and adaptability.
- Hierarchical reasoning design: The two-stage CoT approach that separates template understanding from context-specific realization is intuitive and well-motivated for the meme generation task.

**Weaknesses:**

- Missing critical citations: The paper lacks important references in humor understanding using LLMs and alignment of subjective humor preferences. Notable omissions include work by Hessel et al. (2023), Zhang et al. (2024), Zhou et al. (2025), Kazemi et al. (2025), Liang et al. (2025), Binsted et al. (2006), and Apte et al. The claim about CoT improving VLM reasoning also needs citation support.
- Insufficient human evaluation details: The paper provides no information about human annotators - recruitment methods, number of annotators, compensation, inter-annotator agreement, or specific instructions given. This is critical for reproducibility and validity of human evaluations.

- Unclear evaluation methodology:

- The Human-Likeness Score methodology is unclear. Is it calculated using Gemini-2.5-flash? What's the rationale for using this specific model as a judge? Has this model been validated as correlating with human preferences?
- Figure 4b claims the ranking "aligns more closely with human judgment" but doesn't provide quantitative evidence for this alignment.


- Qualitative vs. quantitative analysis: Figure 6's analysis about Qwen preferring object mentions while Keye prefers human-like states appears purely qualitative. Quantitative metrics supporting these observations would strengthen the claims.
Technical presentation issues:

- Typo: "tives. Concurrently, Ramesh et al. Ramesh et al. (2021)"
- Typo: "Template images of each rannking dataset" (Figure 8)

**Questions:**

1. **Human evaluation protocol**: Can you provide detailed information about your human annotation setup? How many annotators participated? What was the inter-rater reliability? How were they compensated? What specific instructions were provided?

2. **VLM-as-judge validation**: Why was Gemini-2.5-flash chosen for the Human-Likeness Score? Have you validated that this model's predictions correlate with actual human judgments? The 91.3% score seems surprisingly high.

3. **Group-wise ranking validation**: How exactly does Figure 4b demonstrate that group-wise ranking aligns better with human judgment? Can you provide correlation coefficients or other quantitative metrics?

4. **Base model analysis**: For the qualitative observations in Figure 6, can you provide quantitative analysis showing the frequency of object-mention preferences vs. state-description preferences across a larger sample?

5. **Generalization**: How well does HUMOR generalize to meme templates not seen during training? Have you tested on completely novel meme formats?

6. **Computational cost**: What is the computational overhead of the multi-path CoT generation compared to direct generation approaches?

---

> ### Author Response · Authors · 2025-11-25
> **Rebuttal by Authors (Part 1/2)**
>
> We sincerely thank you for your review, your recognition of our work, and your valuable comments. We kindly invite you to review our **General Response** for an overview. Below, we provide a point-by-point response to each of your concerns, detailing the revisions and additional experiments we have conducted to address them.
>
> **1: Missing critical citations**
>
> We appreciate the reviewer’s suggestion to strengthen our literature review. In the revised manuscript, we have expanded the Related Work section to explicitly situate our approach within relevant foundational and recent research. Specifically, we now include citations to seminal humor theory (e.g., Apte; Binsted), recent advances in subjective preference alignment (e.g., Zhang et al., 2024; Zhou et al., 2025), and multimodal CoT reasoning (e.g., Zhang et al., 2023). These additions ensure our contributions are clearly contextualized and well-supported by the broader research landscape.
>
> Regarding the reference to **Kazemi et al. (2025)**  you mentioned, we have conducted a thorough search but were unable to locate this specific work. We did identify a paper titled "SpaceVLM: Sub-Space Modeling of Negation in Vision-Language Models", though its focus appears to be on improvements in CLIP training, which does not seem directly related to the scope of our study. If possible, we would be grateful if the reviewer could share additional details—such as the full title or venue—so that we may appropriately examine and, if relevant, incorporate this citation. We highly value your input and wish to ensure our references are both complete and accurate.
>
> **2: Insufficient human evaluation details and protocol**
>
> We thank the reviewer for highlighting the need for greater clarity regarding our human evaluation process. We appreciate the reviewer’s concern and have revised the paper to make these details explicit so that our human evaluations are fully transparent and reproducible. In response, we have provided comprehensive statements as follows:
>   - Recruitment & Demographics: All annotators were recruited via Prolific under a formal research study. For both the absolute scoring (30 participants, Table 1) and the group-wise ranking study (100 participants, Table 2), we required participants to be native English speakers with a Prolific approval rate of ≥70% to ensure high-quality and culturally grounded meme judgments.
>
>   - Evaluation Methodology:
>     - For the ranking study (Table 2), we employed MaxDiff (Best–Worst Scaling) rather than simple Likert scales. Annotators were repeatedly presented with small groups of memes based on the same template and asked to select the funniest and least funny in each set. This method is known to mitigate scale bias and yield more discriminative preference signals.
>     - For absolute scoring (Table 1), 30 raters evaluated each meme across predefined axes—such as humor, readability, relevance, and originality—using detailed written instructions and examples.
>     - Additionally, we have conducted a validation study (N = 9) demonstrating strong correlation between human-derived rankings and the VLM-based proxies used in Figure 4(b) of the original paper, we also add some additional analysis in Appendix L.3 and L.4 to show more details. Appendix L.3 analyzes human annotator consistency and reveals that human judgments of meme authenticity are extremely unstable. While Appendix L.4 validates group-wise ranking reliability: Gemini’s relative rankings exhibit strong human alignment. These parts both further support the consistency of our evaluation pipeline.
>   - Compensation: All participants were compensated at approximately £9/hour, in accordance with Prolific’s fair payment guidelines.
>
> **3. The Human-Likeness Score (VLM-as-judge validation)**
>
> To address your concern, we verify that the Human-Likeness Score is calculated using Gemini-2.5-pro. In the revision, to further substantiate this choice, we performed systematic benchmarking of several candidate evaluators (Appendix L.1). Gemini-2.5-pro exhibits highest discriminative accuracy, lowest false positive rate, and better statistical robustness than others. These results validate our selection of Gemini-2.5-pro as a reliable evaluator for human-likeness. Below is the concrete benchmark table:
>
> | Evaluator | AUC | Youden_Index | TP | FP |
> |--------|-----|--------------|----|----|
> | Gemini-2.5-pro | 0.7212 | 0.4347 | 0.4680 | 0.0333 |
> | Qwen2.5-32B-Instruct | 0.6082 | 0.1780 | 0.7280 | 0.5500 |
> | GLM-4.1V-9B-Thinking | 0.5629 | 0.1300 | 0.6000 | 0.4700 |
> | Keye-VL-8B-preview | 0.5175 | 0.0387 | 0.0520 | 0.0133 |
> | Qwen2.5-7B-Instruct | 0.5170 | 0.0340 | 0.0440 | 0.0100 |
> | InternVL3-8B | 0.4805 | 0.0313 | 0.9480 | 0.9167 |

---

> ### Author Response · Authors · 2025-11-25
> **Rebuttal by Authors (Part 2/2)**
>
> ------Thank you for continuing to read!------
>
> **4. Figure 4(b) and group-wise ranking validation**
>
> We thank the reviewer for raising this point. To validate the group-wise ranking approach used in Figure 4(b), we conducted an additional human study that directly mirrors the VLM evaluation protocol. For five representative image groups (30 memes in total), nine human annotators ranked six variants—HUMOR-CoT, three CoT baselines, In-the-wild, and Text-Free—using the same criteria rubric employed by Gemini-2.5-pro. We then computed rank correlation between VLM and human preferences. The results show strong alignment:
>   - Mean Spearman correlation: 0.7188 ± 0.2154
>   - Mean Kendall’s τ: 0.6320 ± 0.2269
>
> Some of the results are shown in the table below：
> | Dimension           | Mean Spearman correlation | Standard deviation | Average Kendall's τ |
> |---------------------|---------------------------|--------------------|----------------------|
> | Humor Effectiveness           | 0.82                      | 0.155              | 0.752                |
> | Image-Caption Relevance       | 0.634                     | 0.268              | 0.544                |
> | Theme Relevance | 0.703                 | 0.212              | 0.6                  |
>
> These findings quantitatively confirm that Gemini-2.5-pro’s group-wise rankings are consistent with human judgment, supporting the validity of the relative evaluation presented in Figure 4b. Full details are provided in the Appendix L.4.
>
> **5. Generalization**
>
> We agree that generalization to unseen meme data distribution and formats is essential for real-world applicability.
> To assess this, we conducted a new experiment using the set of meme templates from Appendix K.2 and K.5, which are completely non-overlapping with the training set.
> New Experiment Setup: We collected 20 templates from a Non-Homologous Meme Set and generated captions using HUMOR-CoT and several major VLM baselines (Qwen2.5-7B, Qwen2.5-32B, GLM-4V, InternVL3-8B, Keye-VL-8B).
> Evaluation: We applied the same group-wise VLM ranking protocol as in Figure 4(b), with Gemini-2.5-pro evaluating all variants across eight criteria.
> Results: As shown in the newly added Figure 5 (unseen meme template ranking results), HUMOR-CoT consistently ranks second only to human-created memes, outperforming all VLM baselines. This demonstrates that our method generalizes robustly to unseen templates, and the ranking trends align with those on the in-distribution test set, indicating that HUMOR’s hierarchical CoT avoids overfitting to template-specific structures. Figure 16 shows the generation of our method compared to humans and other VLMs on two specific unseen templates. In Figure 19, we also report more cases in real-world scenario: workplace meme generation.
>
> **6. Computation cost**
>
> For this part, the primary additional cost of our multi-path CoT approach stems from the longer reasoning sequences generated during inference. Empirically, our method requires approximately 2–3× more total output tokens compared to direct generation approaches. While this does increase computational demands during inference, we consider this a worthwhile trade-off given the substantial improvements in humor diversity and quality demonstrated in Table 1. The extended reasoning process enables the model to explore multiple humorous angles and perform more nuanced template-caption alignment, which we have shown to be critical for generating higher-quality memes.
>
> **7. Others**
>
> For other issues like typos and poor representations, we have carefully addressed each of these throughout the revised manuscript to improve clarity and overall text quality. Should any further issues come to mind, we warmly welcome your additional feedback. We would also be grateful to learn which key concerns would most influence your final assessment, as your guidance has been invaluable in strengthening this work. Once again, we deeply appreciate your thoughtful review and the constructive comments that have helped us significantly improve the paper.

---

### Official Review · Reviewer_iDgE · 2025-11-03

**Soundness:** 3
**Presentation:** 4
**Contribution:** 3
**Rating:** 8
**Confidence:** 4

**Summary:**

The paper describes a new framework for generating memes, separating the reasoning from "realization" (interesting choice of words there). The HUMOR framework introduces a hierarchical multi-path CoT base approach to first infer a meme template's approach, ground it in a scene and finally anchor to one (CoT) path using the ground-truth caption.

To do this, they collected a large meme dataset, in-painted it to remove the text, used OCR to extract the text.  They also collected preference data within groups - (groups consist of the same template/punchline schema/topic) from human annotators, learn a pairwise reward model, and aggregating pairwise preferences (using EBC, should account for missing pairs I believe), and then finetunes a meme-generator using group-wise RL, trying to prioritize the higher ranked captions, while trying not to diverge too far from the reference policy.

The experiments and results show better human ranking of the finetuned model outputs over baselines. They also justify the need for group-wise evaluation by showing that using a VLM for evaluation directly is not reliable.

**Strengths:**

Really good framing of the problem - a hard problem to tackle (subjective, multimodal) but a good formulation. Breaking out the chain of thought to first reason about the template and then about the grounding / text is a great way to approach it, similar to how a human would go about making a meme. I like that the authors headed off questions about necessity of using group-wise ranking by showing that absolute VLM scoring is not as aligned with human ranking as group-wise.
Overall - I think taking on this open ended, subjective problem statement and showing a reasonably strong mathematical framework for measuring humor + improving on existing approaches makes this paper a strong candidate.

**Weaknesses:**

There aren't as many weaknesses that I caught in this paper. Theoretically, it makes sense that using GRPO would limit the deviation from the supervised policy, but I wonder how much it ensures the improvement in the generated outputs.

One thing I would note is that a lot of the structures used to improve alignment to human preferences are engineered for this particular scenario of maximizing humor in the generated image. It would be great to show how this could be generalized for other use cases - for example, if we were to use this framework to generate better posters for presentations, we would need different types of pairwise data.

A framework to quantify *how* the types of pairwise data were chosen would be really helpful here - but I also recognize that this comment might be a bit of a nitpick.

The lack of details about the human annotator demographics is slightly concerning - since the results are mostly empirical, that determines if there is actual improvement in humor, or some sort of overfitting happening.

**Questions:**

I appreciate the amount of experiments run, so I am loathe to suggest more ablations. Several of the metrics reported rely on a single language model as a judge - (gemini/keye). Have you considered evaluating with multiple judges? Or are these the judges that did the best/most consistent?

Why are we using bge-base-zh for text similarity? Is a large amount of the corpus in Chinese?

On data collection: 80,000 memes were collected but table 3 in the appendix shows only 3,713? Was this some sort of filtered subset?

On CoT anchoring: how do you select the anchor path in Stage 2? Are results sensitive to the anchoring?

How do you account for offensive memes?

Humor is really subjective - so a lot of the human alignment results the paper demonstrates hinges on the annotator demographics. Do we have a varied group of annotators?

Another thing that your formulation guarantees is that we don't get significantly worse than the supervised policy (by virtue of the KL divergence constraint on the objective). But this does not really show that the performance as judged by humans is getting better, which hinges on having a good human annotator group. Knowing number of raters, how they were asked to rate the images, how strong human-human agreement was would help ground the empirical results better.

**Details Of Ethics Concerns:**

No real ethics concerns, but the results may be biased based on the human annotator demographics.

---

> ### Author Response · Authors · 2025-11-26
> **Rebuttal by Authors (Part 1/3)**
>
> We sincerely thank the reviewer for the thoughtful review and constructive feedback. We kindly invite you to review our General Response for an overview of the key updates. Below, we provide a point-by-point response to each of your concerns, detailing the specific revisions and additional experiments we have conducted to address them.
>
> **1. Generalization to other use cases**
>
> Thank you for the thoughtful suggestion. We agree that the HUMOR framework is not limited to memes and can naturally extend to other open-ended multimodal generation tasks. Conceptually, HUMOR only assumes that (i) **outputs are comparable within coherent groups**, and (ii) **human preferences are more stable in these local comparisons** (as discussed in Section 3). For a task such as presentation-poster generation, a “group” could be defined as posters sharing the same core content but differing in layout, visual style, or typography. Within such groups, pairwise comparisons allow the reward model to focus on aesthetic and usability preferences without being confounded by content.
>
> In terms of data construction, the same recipe we use for memes can be reused in other domains. As detailed in Appendix E, we build a tiered difficulty curriculum of pairwise data (5 datasets from 3 tiers of difficulty), where each tier is guaranteed to have a correct ordering but becomes progressively harder. For posters, the tiers would be instantiated by systematically violating different constraints at increasing granularity (e.g., obvious layout breakdown). Pairs sampled across tiers and upweighted on harder tiers provide a supervision signal that is both confident and discriminative.
>
> **2. Quantification of pairwise data construction**
>
> Thank you for raising this point. We emphasize that our current data construction methodology is already rigorous and sufficient, follows a simple principle: we generate pairs **across multiple difficulty tiers** (from coarse structural violations to subtle near-ties) to ensure that **the reward model receives preference signals of varying granularity**. More details are shown in Appendix E.
> To further formalize the sufficiency of this established process, we have also considered a more explicit selection criterion—monitoring rank-consistency convergence (e.g., Kendall’s τ on held-out groups) to identify which tiers meaningfully contribute to learning. Data collection for a specific difficulty tier is deemed sufficient when the rank consistency plateaus, indicating that the Reward Model has successfully learned the preference boundary at that granularity. While this idea was not included in the current manuscriptdraft, we agree it provides a practical and quantifiable guideline.
>
> **3. Insufficient human evaluation details and protocol**
>
> We thank the reviewer for highlighting the need for greater clarity regarding our human evaluation process. We appreciate the reviewer’s concern and have revised the paper to make these details explicit, ensuring our human evaluations are fully transparent and reproducible. We address the specifics of our protocol in three key aspects:
>
> - **Evaluator Data (Recruitment & Demographics)** All annotators were recruited via Prolific under a formal research study framework. To ensure high-quality and culturally grounded meme judgments, we enforced strict filtering criteria: participants must be Native English Speakers with a Prolific approval rate of $\ge$70%.
>     - Scale: We conducted Absolute Scoring with 30 participants (Table 1) and scaled the key Group-wise Ranking study to 100 distinct participants (Table 2).
>     - Compensation: All participants were compensated at approximately £9/hour, strictly adhering to Prolific’s fair payment guidelines.
>
> - **Diversity Assurance** We agree that humor is subjective, and demographic diversity is vital for ensuring our results reflect a broad consensus rather than niche preferences. By leveraging Prolific’s heterogeneous participant pool and leaving age/gender parameters unrestricted, we captured a naturally diverse demographic spread. Crucially, for the key group-wise ranking (Table 2), aggregating preferences from 100 distinct participants effectively smooths out individual subjectivity.
>
> - **Reliability Assurance** We concur that the validity of our results hinges on the quality of the preference signal. We ensure reliability through rigorous methodology and validation:
>     - Methodology (MaxDiff): For the ranking study, we employed **MaxDiff (Best–Worst Scaling)** rather than simple Likert scales. This method forces explicit trade-offs and is theoretically proven to mitigate individual scale biases.
>     - Statistical Evidence: The reliability of these judgments is evidenced by the **strong statistical significance** of our results (often $p < 0.001$ in Table 2). Such clear separation between methods would be statistically impossible if human-human agreement were low or dominated by idiosyncratic noise.

---

> ### Author Response · Authors · 2025-11-26
> **Rebuttal by Authors (Part 2/3)**
>
> **4. Rationale for VLM Judge Selection**
>
> We agree that relying on specific models requires rigorous justification. Our systematic benchmarking confirms that Gemini-2.5-pro was selected for its superior discriminative precision (highest AUC of 0.7212 and lowest FPR of 3.33%) and its ability to provide a more objective signal than human annotators, who exhibited significant instability in distinguishing AI-generated content. Please refer to **Point 3 of the General Response** for the detailed experimental results and validation analysis.
>
> While Gemini serves as the ideal judge for final discrimination, the internal reward signal for RL optimization requires a different set of capabilities—strict ranking consistency. Distinct from the evaluator role, we selected Keye-VL as the content judge in reward scoring based on its score monotonicity and stability (verified in Appendix F.2). To validate this, we constructed caption groups with controlled quality levels (ranging from 0 to 50). In systematic comparisons, Keye-VL exhibited the clearest monotonic trend, where its scores consistently increased alongside intended quality levels(Table 3). In contrast, other candidates (e.g., Qwen-based scoring or raw logits) showed weaker correlation or significant noise. Furthermore, Keye-VL demonstrated superior cross-distribution generalization, remaining stable even when scoring content generated by other models. This rank consistency makes it the most reliable supervisor for our RL optimization.
>
> **5. Clarification on Text Similarity Metric (bge-base-zh)**
>
> We sincerely thank the reviewer for their keen attention to detail. We acknowledge that the use of bge-base-zh was a configuration oversight. Although the model supports multilingual encoding, we agree that an English-optimized model is strictly more appropriate for our dataset.
> We have re-computed the Similarity metrics in Table 1 using the English-optimized bge-base-en-v1.5. While the absolute values shifted slightly, the relative performance trends and model rankings remain entirely consistent with our original submission, with HUMOR continuing to outperform all baselines. We have updated these figures in the revised manuscript to ensure rigorous accuracy.
>
> **6. Clarification on Dataset Size Discrepancy**
>
> We appreciate the opportunity to clarify the data processing pipeline. The 80,000 figure refers to the raw corpus collected from public sources. Table 4, however, reports the curated instruction-tuning dataset specifically used for model training.
> To ensure high-quality generation, we applied a rigorous filtering process:
> - **Label Availability**: From the full corpus, we first retained memes with complete human-evaluation meta-labels (essential for our intent and emotion modeling), yielding roughly 4k candidates.
> - **Safety & Quality**: Consistent with our Ethics Statement, we further filtered this subset to remove low-resolution images and content containing violence or toxicity.
>
> This resulted in the final set of 3,713 high-quality samples used for SFT. Our design deliberately prioritizes data quality and controllability over raw scale to prevent the model from learning toxic patterns or noise. In the revised manuscript, we have updated the dataset description to explicitly specify that the reported statistics refer strictly to this final SFT dataset, ensuring there is no ambiguity regarding the training data size.

---

> ### Author Response · Authors · 2025-11-26
> **Rebuttal by Authors (Part 3/3)**
>
> **7. Clarification on CoT Anchoring and Sensitivity**
>
> We appreciate the opportunity to clarify the anchoring mechanism and its stability. In training, we do not "select" a path from model hypotheses. Instead, the anchor path is **reverse-engineered directly from the ground-truth caption**. By inferring the reasoning trace from the target intent, we ensure the anchor always corresponds to a valid, high-quality realization rather than an uncertain model hypothesis. While in inference, the anchor is determined strictly by the input context (e.g., user intent or scene labels) and constrains the reasoning module. To prevent future confusion, we have revised Section 4.1 in the updated manuscript to explicitly separate and delineate the descriptions of the training and inference workflows, ensuring this distinction is structurally clear and easy to follow.
>
> For the sensitivity as steerability, the model is indeed responsive to the anchor, which we regard as a desirable feature reflecting steerability. Our "Distance under Context Swap" metric in Table 1 quantitatively validates this: HUMOR produces substantially different outputs (High Distance: 0.590) when the context anchor is swapped. This confirms that the model does not ignore the anchor but reliably adapts its reasoning path to match the specified intent.
>
> **8. Response to Offensive Content Concerns**
>
> We acknowledge the critical importance of addressing potentially offensive content and have incorporated safety mechanisms at multiple stages. First, during data preprocessing, we rigorously filtered out memes containing violence, hate speech, or other toxic content, with specific protocols detailed in Appendix B of the revision. Crucially, our Hierarchical CoT explicitly models creation intent—distinguishing between tags such as "offense" versus "entertainment"—which allows us to steer generation away from offensive interpretations via control prompts, effectively treating safety as a controllable attribute rather than an uncontrollable side effect. Finally, for real-world deployment, we recommend combining HUMOR with standard safety guardrails (e.g., content filters) to mitigate any residual risk in open-ended generation settings.
>
> We once again express our sincere gratitude for the reviewer’s constructive critique, which has significantly elevated the rigor and clarity of our manuscript. We remain fully open to further dialogue and would welcome any additional questions or feedback during the discussion period to help us refine this work even further.

---

### Author Response · Authors · 2025-11-25
**General Response (Part 1/2)**

We sincerely thank all reviewers for their insightful comments and constructive suggestions. We are honored that the reviewers generally recognize the problem formulation of our work (Reviewer iDgE, Reviewer VisL, Reviewer ftT4), math framework reasonableness (Reviewer iDgE, Reviewer VisL, Reviewer uciU, Reviewer ftT4), and comprehensive experiments (Reviewer iDgE, Reviewer VisL, Reviewer uciU, Reviewer ftT4). In response to your feedback, we have carefully revised the manuscript and supplemented additional information throughout the main paper and appendices to better explain our methodology, experiments, and contributions. The revised version of paper is updated on openreview, all modifications in the manuscript have been highlighted in blue text. Below we summarize the key revisions made:

1. **Improved Framework Clarity (Sec. 4.1):** We have partially rewritten the framework section to more clearly distinguish between the training and inference procedures. The hierarchical CoT pipeline is now described in a step-wise manner, emphasizing the two-stage supervision process. And we want to emphasize that we have already introduced a visual dataflow diagram (Figure 2) to illustrate the path exploration and anchoring mechanisms in the original paper.

2. **Expanded Ablation Study (Sec. 5.1):** To provide a deeper understanding of our method’s advantages, we have enriched the ablation study with detailed textual explanations comparing HUMOR-CoT against three ablation variants. Although these results were originally included in Table 1 (Fine-tuned models), the corresponding analysis was initially lacking. The updated section now elaborates on how HUMOR-CoT consistently achieves higher performance in humor, relevance, readability, and diversity, and explains why alternative CoT strategies underperform—for instance, due to over-decomposition or overly conservative stylistic choices.

3. **Comprehensive Dataset Analysis (Appendix A):** We have added a thorough characterization of both the training and evaluation datasets. This includes:
    - Appendix A.1: Distributions of meme templates and semantic categories.
    - Appendix A.2: Empirical analysis of the Context-Swap Distance metric (used in Sec. 5.1), supported by histograms and boxplots.
This expansion helps clarify the intrinsic semantic density across meme variations and explains why the observed distances typically fall within the 0.5–0.6 range—a point raised by the reviewer uciU.


4. **Clarified Content Reward Design (Appendix F.2):** Since the content reward is not the core innovation of our method—serving rather as an auxiliary component in RL training—we didn't mention details in the manuscript. In this revision, we have added a dedicated subsection to justify its design. This includes: the choice of evaluator(judger) VLM, tests for its reliability and consistency, and a comparative analysis to ensure reward stability across different templates. These additions directly address reviewer concerns regarding reward validity and implementation.

5. **Added Pseudocode for EBC (Appendix G):** To enhance reproducibility, we also have included full pseudocode (Algorithm 1) for the Expected Borda Count (EBC) mechanism, detailing the ranking aggregation and probability integration processes used in our reward model and RL pipeline.

6. **Supplementary Experimental Results (Appendix J):** We also have expanded the experimental results with the following additions:
    - J.1: VLM Reclassification Results—A new experiment where a strong VLM classifier re-evaluates generated captions across four semantic axes (emotion, intention, theme, style). Results in Table 5 show that HUMOR-CoT outperforms both 7B and 32B base models in semantic faithfulness.
    - J.2: Human Ranking of Reward-Model Dataset—We provide MaxDiff-based human ranking results for all meme groups used in Sec. 5.3 (Fig. 14), validating the reliability of group-wise human supervision.

7. **Additional Qualitative & Diagnostic Analyses (Appendix K)**: According to the suggestions of reviewers, we includes extensive qualitative evidence supporting our main findings:
    - K.1: Side-by-side comparisons of HUMOR-CoT against other CoT strategies across multiple user-intent clusters (e.g., romance, Christmas, family tradition, delayed surprise).
    - K.2: Generalization tests on unseen (OOD) templates, showing HUMOR-CoT ranks second only to human-created memes across 20 new templates.
    - K.3: Identification of risk cases, such as ambiguous cultural humor or risky metaphors.
    - K.4: Failure-case diagnostics analyzing where reasoning paths break down or visual grounding fails.
    - K.5: Extension tests on real-world applications, including several workspace meme generation results with single/multi-panel and  with different emotional/contextual tags.

---

### Author Response · Authors · 2025-11-25
**General Response (Part 2/2)**

------Thank you for continuing to read!------

8. **Evaluator Reliability and Ranking Stability (Appendix L)**: This section provides a comprehensive validation of the Gemini-2.5-pro evaluator used in our pipeline.
    - L.1 benchmarks six VLM candidates and shows that Gemini uniquely offers the highest AUC and near-perfect specificity, preventing evaluator-induced penalties on human memes.
    - L.2 establishes statistical significance and reproducibility of Human Rate differences via two-proportion tests and subset-resampling.
    - L.3 analyzes human annotator consistency and reveals that human judgments of meme authenticity are extremely unstable (very low κ), reflecting the inherent difficulty of this task. In contrast, Gemini’s continuous scores align with ground-truth authenticity substantially better than human annotators, showing that in this setting Gemini is more faithful to true labels than humans themselves.
    - L.4 validates group-wise ranking reliability: Gemini’s relative rankings exhibit strong human alignment (Spearman 0.72, Kendall τ 0.63) across diverse meme sets.

    Together, these analyses confirm that Gemini-2.5-pro provides a stable, human-aligned, and bias-controlled evaluation basis for both absolute Human Rate and relative ranking comparisons.

9. **Full Prompts for Reproducibility (Appendix M):** To facilitate replication of our results, we now provide all prompts used in the pipeline in Appendix M. We will also open-source all the training and evaluation codes.

10. **Strengthened Ethical Statement:** We have updated the **Ethical Statement** in the revised paper to clarify that:
    - all datasets are publicly released after filtering sensitive information, under appropriate research licenses;
    - human annotators were recruited via legitimate platforms and fairly compensated;
    - informed consent was obtained prior to surveys;
    - harmful content was filtered during preprocessing;
    - and standard safety measures are recommended for deployment.


Through substantial revisions across the main paper and appendices A–M, we have incorporated new analyses, clarifications, and experiments to directly address the reviewers' valuable concerns regarding clarity, evaluation fairness, and reproducibility. We believe these additions have significantly strengthened the manuscript and addressed the raised points. We are deeply grateful for the insightful feedback and welcome any further questions or suggestions.

---

### Note · Program_Chairs · 2026-01-17
**Submission Desk Rejected by Program Chairs**

The following references in this submission do not refer to real documents and/or have major errors in bibliographic information:

 Jack Hessel, Ari Holtzman, Maxwell Forbes, Ronan Le Bras, and Yejin Choi. Winning the New Yorker caption contest with a Scooby-Doo clone. In Proceedings of the IEEE/CVF Conference on Computer Vision and Pattern Recognition (CVPR), pp. 2653-2664. IEEE, 2023a.